# MAMBA: an Effective World Model Approach for Meta-Reinforcement Learning

**Zohar Rimon**[1*]    **Tom Jurgenson**[1*]    **Orr Krupnik**[1]    **Gilad Adler**[2]    **Aviv Tamar**[1]

[1]Technion - Israel Institute of Technology  [2]Ford Research Center Israel

## Abstract

Meta-reinforcement learning (meta-RL) is a promising framework for tackling challenging domains requiring efficient exploration. Existing meta-RL algorithms are characterized by low sample efficiency, and mostly focus on low-dimensional task distributions. In parallel, model-based RL methods have been successful in solving partially observable MDPs, of which meta-RL is a special case. In this work, we leverage this success and propose a new model-based approach to meta-RL, based on elements from existing state-of-the-art model-based and meta-RL methods. We demonstrate the effectiveness of our approach on common meta-RL benchmark domains, attaining greater return with better sample efficiency (up to $15\times$) while requiring very little hyperparameter tuning. In addition, we validate our approach on a slate of more challenging, higher-dimensional domains, taking a step towards real-world generalizing agents.

## 1 Introduction

There is a growing interest in applying reinforcement learning (RL) agents to real-world domains. One requirement of such applications is for the agent to be able to generalize and solve scenarios beyond its training data. Meta-reinforcement learning (meta-RL) is a line of work aiming to allow RL agents to generalize to novel task instances, attempting to learn a general control policy over a family of tasks. The term meta-RL covers a wide range of approaches (Duan et al., 2016; Finn et al., 2017; Zintgraf et al., 2019) which have produced increasingly impressive results in recent years.

One line of work aiming to solve meta-RL problems is *context-based* meta-RL (Rakelly et al., 2019; Zintgraf et al., 2019; 2021). These algorithms encode trajectories of training tasks in a *context* or *belief* variable, and infer it from data at test time in order to identify the task at hand and solve it. This approach is Bayesian in nature, attempting to balance exploration (to identify the task parameters) and exploitation (to succeed in the current task). While state-of-the-art results have been reported on simulated robotic domains (Zintgraf et al., 2021; Hiraoka et al., 2021), previous meta-RL methods tend to be sample inefficient, as they rely on model-free RL algorithms. In addition, they have mostly been shown to succeed only in low-dimensional task distributions.

Model-based RL is known to be more sample-efficient than its model-free counterpart (Janner et al., 2019; Ye et al., 2021). By their nature, model-based RL algorithms may also be more flexible, as they learn a model of the environment and use it to plan or learn policies, instead of learning control policies by directly interacting with the environment. A recent line of work in model-based RL proposed Dreamer, an approach which has shown versatility to a variety of domains (Hafner et al., 2019a; 2020; 2023). Dreamer has delivered the sought-after sample efficiency of model-based RL on a variety of domains, while requiring very little hyperparameter tuning. In addition, due to its recurrent latent encoding of trajectories, it has been successful in solving partially observable Markov decision process (POMDP) scenarios (Hafner et al., 2020; 2023). POMDPs can be viewed as a general case of the context-based meta-RL formulation (where the context identifying the task is unobserved by the agent). In light of its favorable properties, we believe that Dreamer is a good fit for meta-RL. Therefore, in this work we propose to use Dreamer to solve meta-RL scenarios. To the best of our knowledge, this is the first use of the Dreamer architecture in meta-RL.

We compare Dreamer to state-of-the-art meta-RL algorithms (Zintgraf et al., 2019; 2021), and propose MAMBA (for **M**et**A**-RL **M**odel-**B**ased **A**lgorithm), an algorithm leveraging the best compo-

---

*Contributed equally, correspondence to zohar.rimon@campus.technion.ac.il and tomjurgenson@gmail.com. Code available at: `https://github.com/zoharri/mamba`.

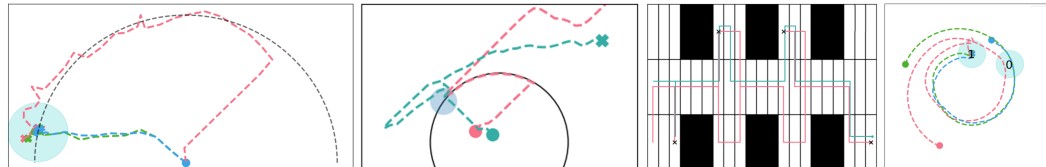

Figure 1: Behaviors learned by MAMBA in various domains. In all of the figures, the first, second and third episodes are shown in red, green and blue, respectively. From left to right: Point Robot Navigation, Escape Room, Rooms-$N$ and Reacher-$N$ (plotted point represents the end-effector; see Sec. 4 for details). As the environments have sparse rewards, MAMBA evidently learns near-Bayes-optimal behavior: it explores the environment in the first episode and exploits the information in the subsequent ones. For additional visualizations see https://sites.google.com/view/mamba-iclr2024.

nents of each. MAMBA outperforms meta-RL and model-based RL baselines on several well-studied meta-RL benchmarks, both in terms of task success and in sample efficiency. In addition, we propose a new class of challenging meta-RL domains with high-dimensional task distributions, which can be decomposed into lower-dimensional sub-tasks. By bounding the number of tasks required to obtain near-Bayes-optimal behavior (based on the analysis of Rimon et al. 2022), we show that the ability to decompose the task into sub-tasks is crucial for solving such domains efficiently. We then empirically show that MAMBA can decompose the problem better than existing meta-RL approaches, which leads to significantly better performance.

To summarize, our main contributions are:

1. We present a shared formulation of context-based meta-RL algorithms and the Dreamer line of work, highlighting their similarities and differences.

2. We develop MAMBA, a sample-efficient meta-RL approach which outperforms a variety of baselines on meta-RL benchmarks, while requiring very little hyperparameter tuning.

3. We highlight an interesting class of meta-RL environments with many degrees of freedom that is difficult for existing meta-RL methods to solve. We analyze these domains theoretically, and demonstrate the ability of MAMBA to solve them efficiently.

## 2 BACKGROUND

In this section we present background on partially observable Markov decision processes (POMDP) and meta-RL (Sec. 2.1). Next, we describe context-based algorithms and model-based RL (Sec. 2.2), which are popular approaches for solving meta-RL and POMDPs respectively.

### 2.1 PROBLEM FORMULATION

**Partially observable Markov decision processes** (POMDP, Cassandra et al. 1994) are sequential decision making processes defined by a tuple $(\mathcal{S}, \mathcal{A}, \rho_0, P, R, T, \Omega, O)$, where $\mathcal{S}$ and $\mathcal{A}$ are the state and action spaces respectively. The process is initialized at a state sampled from the initial state distribution $s \sim \rho_0$ and evolves according to the Markovian state transition probability $P(s'|s,a)$ and reward function $R(r|s,a)$ for $T$ time steps. The states are unobserved by the agent, which has access to observations from the observation space $\Omega$, sampled at each time step from $O(o|s)$. Let $\tau_t = (o_0, a_1, r_1, o_1, \ldots a_t, r_t, o_t)$ be the history up to time step $t$. The objective is to learn a policy $\pi(a_{t+1}|\tau_t)$ which maximizes the expected return $J^\pi = \mathbb{E}_{\pi,P,O}\left[\sum_{t=1}^{T} r_t\right]$, where $\mathbb{E}_{\pi,P,O}$ denotes the expectation with respect to the policy, observation function and POMDP transitions.

In **meta-reinforcement learning (meta-RL)** we consider a distribution $p(M)$ over a space of MDPs $\mathcal{M}$, where every MDP is a tuple $(\mathcal{S}, \mathcal{A}, \rho_0, P_M, R_M, T, K)$. Here $\mathcal{S}$ and $\mathcal{A}$ are the state and action spaces respectively, shared among all MDPs, while $P_M(s'|s,a)$ and $R_M(r|s,a)$ are the MDP-specific transition and reward functions. The agent plays $K$ consecutive episodes in the same MDP, each consisting of $T$ time steps. The full horizon $H = K \cdot T$ comprises the *meta-episode*, with each of its $K$ sub-trajectories denoted *sub-episodes*. The objective is to learn a policy $\pi(a_{t+1}|\tau_t)$

which maximizes the expected return $J^\pi = \mathbb{E}_{M \sim p}\left[\mathbb{E}_{\pi,M}\left[\sum_{t=1}^{H} r_t\right]\right]$. Meta-RL can be seen as a special case of a POMDP, where the unobserved components are $P_M$ and $R_M$, and the transitions are *episodic* – every $T$ time steps the state is reset.

**Bayes-Adaptive MDP:** Solving POMDPs is known to be intractable (Papadimitriou & Tsitsiklis, 1987). While an optimal policy is generally history-dependent, a convenient representation of the optimal policy uses a *sufficient summary* of the history (also called *sufficient statistic* or *information state*; Subramanian & Mahajan 2019) One such representation is the *belief*, $b_t = p(R, P|\tau_t)$, the posterior probability over $R, P$ given the history of transitions up to time $t$. The POMDP can be cast as an MDP over the belief space, known as a Bayes-adaptive MDPs (BAMDP, Duff 2002). In BAMDP, the state $s_t$ is augmented with the belief $b_t$ and is denoted as the augmented state $s_t^+ = (s_t, b_t)$. Formally, the BAMDP is a tuple $(S^+, A, \rho_0^+, P^+, R^+, T)$, where $S^+$ is the space of augmented states, $A$ is the original action space, and $\rho_0^+$, $P^+$, and $R^+$ operate over augmented states. An optimal policy $\pi_{BO} \in \arg\max_\pi J^\pi$ is termed *Bayes-optimal*, and is history-dependent in the general case.

## 2.2 Context-Based Meta-RL and Model-Based Algorithms for POMDP

**Context-based meta-RL:** One popular approach to solving meta-RL is to encode the observed history $\tau_t$ into a latent representation which approximates the belief at time $t$, $b_t$, and learn a belief-conditioned policy in order to maximize the return $J^\pi$.

A recently successful context-based algorithm, VariBAD (Zintgraf et al., 2019), learns $b_t$ using a Variational Autoencoder (VAE, Kingma & Welling 2013): a recurrent neural network (RNN) encoder encodes $\tau_t$ into the latent space of the VAE which is interpreted as $b_t$. Unlike other VAE formulations which only reconstruct the input (namely $\tau_t$), VariBAD reconstructs the full trajectory $\tau_T$ from $b_t$. Meanwhile, a policy is trained on the augmented state $(s_t, z_t)$ using a model-free RL algorithm (specifically PPO, Schulman et al. 2017). For stability, the gradients do not flow from the RL objective through $z_t$ into the encoder, i.e., the belief is shaped **only** by the reconstruction signal.

**Recurrent latent model-based algorithms for POMDP:** Model-based RL (MBRL) algorithms model the dynamics $P$ and rewards $R$ given data, and then use the learned models $\hat{P}$ and $\hat{R}$ to imagine future outcomes and optimize a policy. This approach has been shown empirically to be very sample-efficient (Schrittwieser et al., 2020; Ye et al., 2021; Hafner et al., 2023). In particular, *latent* MBRL algorithms are an appealing approach to handle partial observability by first encoding a sequence of previous observations $\tau_t$ into a latent space $z_t \in Z$, and learning dynamics $\hat{P}(z_{t+1}|z_t, a_t)$ and rewards $\hat{R}(r_t|z_t, a_t)$ directly in the latent space. This approach offers two main advantages: first, by encoding sequences of observations we allow $z_t$ to recall information from previous time steps which might help with the partial observability aspect. Second, imagined rollouts used to optimize the policy are defined over a much smaller space ($|Z| << |\Omega|$), allowing more efficient planning.

One such family of algorithms is Dreamer (Hafner et al., 2019a; 2020; 2023). In Dreamer, the world model $w$ is a type of RNN dubbed *recurrent state space model* (RSSM, Hafner et al. 2019b), which predicts observable future outcomes from sequences of past observations. Concretely, a random sequence of transitions of length $L$, $\tau_{i:i+L}$, is encoded into a latent space $z_{i:i+L}$ using the RSSM. Then, for every $t \in [i, \ldots, i + L]$, Dreamer predicts from $z_t$ the next reward $r_{t+1}$ and observation $o_{t+1}$ and minimizes $\mathcal{L}_{pred} = -\log w(r_{t+1}|z_t) - \log w(o_{t+1}|z_t)$.

In parallel, Dreamer trains a policy conditioned on $z_t$ which maximizes the returns of imagined future latent-space trajectories of length $N_{IMG}$. Finally, as in VariBAD, gradients from the policy do not propagate through to the world model $w$.

## 3 Method

In this section we present MAMBA. We start by motivating two main reasons for our selection of Dreamer (Hafner et al., 2023) as the backbone of our method. First, in Sec. 3.1, we demonstrate that model-based algorithms and Dreamer in particular are in fact structurally similar to state-of-the-art

meta-RL algorithms. Then, in Sec. 3.2 we explain why in some meta-RL tasks the world model learning in Dreamer is advantageous. Finally, we present MAMBA in Sec. 3.3.

## 3.1 COMPARING VARIBAD AND DREAMER

VariBAD is a meta-RL algorithm, while Dreamer is a model-based RL method intended to solve POMDPs. Despite that, they share many similar attributes. We next describe the commonalities and differences between them.

At the core of both algorithms lie two fundamental components: a recurrent predictor for observable outcomes and a latent-conditioned policy. The *recurrent predictor* is realized by the belief encoder in VariBAD and by the world model in Dreamer. Both encode a sequence of observations into a latent space representation $z_t$ using a RNN structure. Then, $z_t$ is used to predict observable outcomes such as future observations and rewards. The *latent-conditioned policy* is trained via online RL objectives by predicting actions given $z_t$.

Despite their similarities, there are two crucial differences between VariBAD and Dreamer. First and foremost, during training, VariBAD uses its recurrent predictor to encode *the full history $\tau_H$* starting at the very first time step of the meta-episode, while Dreamer encodes sub-trajectories that do not necessarily start at the first time step. From a computational standpoint, this makes Dreamer less computationally demanding. However, in meta-RL scenarios it could create unnecessary ambiguity as hints regarding the identity of the MDP (namely $P_M$ and $R_M$), collected during time steps predating the sub-trajectory, are ignored. We refer to this as the *encoding context*.

A second key difference is the recurrent predictor reconstruction window. In Dreamer, the world model performs *local reconstruction* (predicts a single step into the future), while VariBAD performs *global reconstruction* (predicts an entire trajectory, past and future, from every $z_t$).

The differences in encoding context and local vs. global reconstruction inspire our algorithmic improvements in MAMBA (see Sec. 3.3). Other technical differences are described in Appendix B.

## 3.2 DECOMPOSABLE TASK DISTRIBUTIONS

In this section we focus on the difference in reconstruction between VariBAD and Dreamer. To this end we consider a specific family of task distributions and explain why they are difficult for meta-RL algorithms to solve, by extending the theoretical analysis of Rimon et al. (2022). We then show that utilizing the decomposability of these tasks makes them amenable to a much more efficient solution. Finally, we relate this decomposability to the local reconstruction horizon, and posit that Dreamer should solve these tasks more effectively. We corroborate this in experiments (Sec. 4).

### 3.2.1 PAC BOUNDS

Meta-RL has been shown to scale badly with respect to the number of degrees of freedom (DoF) in the task distribution (Mandi et al., 2022). Rimon et al. (2022) devised PAC (Probably Approximately Correct) upper bounds on the Bayes-suboptimality with respect to the number of training tasks. They showed an exponential dependency on the number of DoF of the task distribution.

Many real-world domains with a high number of DoF can be decomposed into sub-tasks, each with a low number of DoF. For example, a robot may be required to solve several independent problems in a sequence. In this section we analyze this family of task distributions, which we term *decomposable task distributions* (DTD), and prove that they have significantly better PAC bounds than the general case. In addition, we demonstrate that on DTD the local reconstruction approach taken by Dreamer is advantageous compared to the global reconstruction in VariBAD.

**Bounds on Bayes-suboptimality for DTD:** For a task that can be decomposed to $K$ independent tasks (up to their initial state distribution), each with $D$ DoF, a naive approach (such as Rimon et al. 2022) would lead to a bound that depends exponentially on $\mathcal{O}(D \times K)$. However, as we demonstrate in the next theorem, a method that takes task independence into account obtains an exponential dependency on $\mathcal{O}(D)$ and a linear dependency on $K$, equivalent to solving each task independently (proofs and full details in Appendices A.1 & A.2). We follow the formulation in Rimon et al. (2022) and assume access to $N$ samples from the prior task distribution $f$. We estimate

$f$ using a density estimation approach, obtaining $\widehat{f}$, and find the Bayes-optimal policy with respect to $\widehat{f}$, i.e. $\pi^*_{\widehat{f}} \in \arg\max J^{\pi}_{\widehat{f}} = \arg\max \mathbb{E}_{M \sim \widehat{f}}\left[\mathbb{E}_{\pi,M}\left[\sum_{t=1}^{H} r_t\right]\right]$.

**Theorem 1** *(informal) Under mild assumptions, there exists a density estimator $\hat{f}$ which maps $N$ samples from a decomposable task distribution to a distribution over the task space, such that with probability at least $1 - 1/N$:*

$$\mathcal{R}(\pi^*_{\widehat{f}}) \leq K \cdot \mathcal{O}\left(\left(\frac{\log N}{N}\right)^{\frac{\alpha}{2\alpha+D}}\right)$$

Where $\mathcal{R}$ is the sub-optimality with respect to the true prior distribution and $\alpha$ is a constant which depends on the smoothness of the prior task distribution.

The key to proving Theorem 1 is to estimate each of the sub-task distributions individually, and not the entire task distribution at once, as was suggested by Rimon et al. 2022 (which leads to bounds with exponential dependency on $D \times K$ instead of $D$). This theoretical result is in line with our hypothesis – *concentrating on the current sub-task (or performing local reconstruction) at each time step is crucial for DTD*.

For example, in the Rooms-$N$ environment (see Sec 4), although the total number of DoF is $N$ (number of rooms), the difficulty of the task does not increase, as the agent should solve each of the rooms separately. In comparison, our theorem states that a standard $N$-dimensional navigation goal, which has the same number of DoF, should be much harder since all DoF must be estimated at once.

### 3.2.2 DECOMPOSABILITY AND LOCAL RECONSTRUCTION

As described in Sec. 3.1, to learn a latent representation, VariBAD utilizes a global reconstruction approach, whereas Dreamer utilizes a local one. Based on Theorem 1, we hypothesize that the latter approach is better for DTD (as the former approach reconstructs multiple sub-tasks at once).

To test our hypothesis, we compare VariBAD with the original reconstruction loss to a variant of VariBAD reconstructing only a small window of 10 transitions around the current time step. We evaluate the two variants on the Rooms-$N$ environment with 4 and 5 rooms, where the agent should explore consecutive rooms, passing to the next room after solving the current one (see Sec. 4 for details). We observe that using the local reconstruction is favorable in terms of total return in both the Rooms-4 and Rooms-5 scenarios (see Sec. 4).

To further analyze this phenomenon, we visualized the reward prediction using the learned decoder in VariBAD in each state of the state space and for every encoded belief (Fig. 2). We observe that VariBAD (without local reconstruction) wrongfully predicts rewards in adjacent states to the agent (Fig. 2 top (b), (f)), and thus the agent stays in the same location without a goal (as indicated by the constant state index). In comparison, introducing local reconstruction makes the **local** reward prediction much more accurate, as indicated in Fig. 2 bottom (d), (e). However, comparing top (f) and bottom (f), local reconstruction hurts the prediction of rewards in distant states. This helps the local reconstruction agent to better focus on the goal and follow near-optimal behavior. For further quantitative analysis see Appendix E. This analysis shows the efficacy of local reconstruction in comparison to a global one on DTD, which motivates our design choices in the next section.

### 3.3 MAMBA: METR-RL MODEL-BASED ALGORITHM

In this section we describe MAMBA, our approach incorporating the model-based Dreamer into the meta-RL settings. We base MAMBA on DreamerV3 (Hafner et al., 2023) since it was shown to be a high-performance, sample-efficient and robust algorithm applicable out-of-the-box to many partially observable scenarios with various input modalities. Moreover, as described in Sec. 3.1 the RSSM (Hafner et al., 2019b) of the Dreamer architecture plays the same role as the *belief encoder* in context-based meta-RL algorithms (Zintgraf et al., 2019; 2021) allowing easy application of tools and conclusions from these algorithms in MAMBA. Finally, the world model in Dreamer learns by reconstructing a single step into the future, a preferable property compared to algorithms which use a long reconstruction horizon (Sec. 3.2). Next, we describe the modifications on top of Dreamer to create MAMBA:

**Modifying trajectories to support model-based meta-RL algorithms:** We augment the observations $o_t$ with the reward and the current time step $o'_t = [o_t, r_t, t]$; both are readily available in any environment and do not require any special assumptions. In meta-RL it is common to incorporate the rewards into the observation (Duan et al., 2016; Rakelly et al., 2019; Zintgraf et al., 2019) as it helps in identifying the sampled MDP. As for adding $t$, it allows to model the reset of the environment after every $T$ steps (see Sec. 2).

**Sampling full meta-episodes:** To avoid missing signals about the identity of the current MDP (see *encoding context* in Sec. 3.1), we modify Dreamer to keep track of all information from the entire meta-episode when encoding $z_t$ using the RSSM. The original Dreamer only encodes $z_t$ for limited random sequences of actions; it samples a trajectory $\tau$ (meta-episode) from the replay buffer, samples a random starting index $t$, and then encodes a fixed-length sequence of $L = 64$ transitions obtaining $z_{t:t+L}$. We emphasize

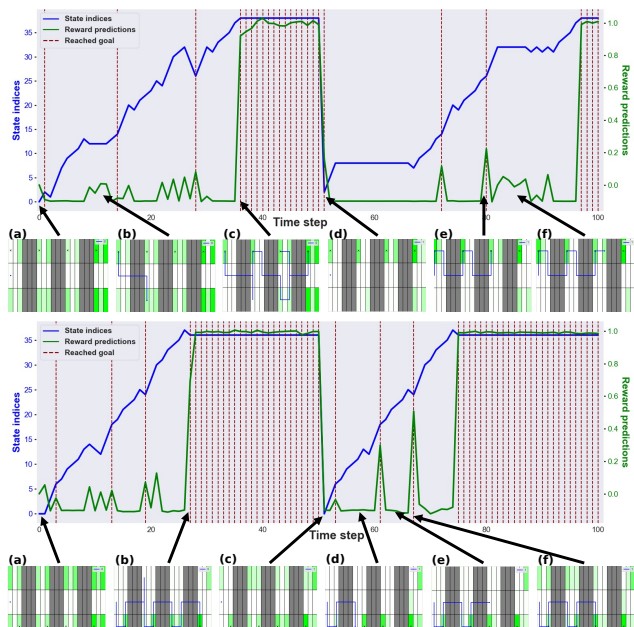

Figure 2: Reward prediction at current location (green), state identifier (blue), and goal state indication (red) through time on the Rooms-4 environment. Top: vanilla VariBAD (global reconstruction), bottom: VariBAD with local reconstruction. At the highlighted time steps (indicated by arrows) we plot the reconstructed reward for every grid cell, where positive reward prediction is represented as green, and the agent trajectory is drawn in blue.

that using this approach, information gathered at steps $1, \ldots, t-1$ is discarded. Instead, in MAMBA, we randomly sample the meta-episode $\tau$, and feed it in its entirety into the RSSM, obtaining $z_{1:H}$. Although this creates a computational burden, as our experiments show, this allows MAMBA to obtain superior performance to Dreamer (see Sec. 4).

To alleviate some of the computational load, after obtaining $z_{1:H}$ when optimizing the RSSM and policy we sample $L$ indices $i_1, \ldots, i_L \sim Uniform(H)$ and optimize (maximize reconstruction in the RSSM, and maximize the return for the policy) only according to $z_{i_1}, \ldots, z_{i_L}$. Formally, let $\mathcal{L}_w(z)$ and $\mathcal{L}_\pi(z)$ be the loss functions for the world model $w$ and the policy $\pi$ from a single latent vector $z$. Then, in Dreamer the world-model loss is $\sum_{t=i}^{i+L} \mathcal{L}_w(z_t)$ and the policy loss is $\sum_{t=i}^{i+L} \mathcal{L}_\pi(z_t)$, while in MAMBA the world-model loss is $\sum_{t=1}^{L} \mathcal{L}_w(z_{i_t})$ and the policy loss is $\sum_{t=1}^{L} \mathcal{L}_\pi(z_{i_t})$. Thus, despite the difference in RSSM input sequence lengths ($L$ in Dreamer and $H$ in MAMBA), during optimization Dreamer and MAMBA are updated according to the same number of latent vectors $z_t$.

**Scheduling the world model horizon:** We found that during the initial update steps, unrolling the world model for the entire meta-episode length $H$ is inaccurate, due to the inaccuracy of the yet-untrained RSSM. In light of this inaccuracy, we reduce the computational load by learning only from prefixes of meta-episodes $\tau_{H'}$ (where $H'$ is the prefix length). As the world model improves, we can increase $H'$ and roll the model out for more steps. We therefore introduce a schedule: let $N_T$ be the total number of environment steps, and define $N_{WM} << N_T$. The world model horizon $H'$ increases linearly from some initial value of $WM_{initial}$ at the start of training to the full meta-episode length $H$ when $N_{WM}$ environment steps are taken.

Although the three modifications from Dreamer to MAMBA may appear technical, they are motivated by the concrete differences between general POMDP and the meta-RL problem. In the following section, we show that they also lead to dramatic performance improvement in meta-RL, over both the original Dreamer and state-of-the-art meta-RL methods.

## 4 EXPERIMENTS

In this section we provide an investigation of the effectiveness of MAMBA. We first show that MAMBA is a high-performing meta-RL algorithm: compared to baselines it obtains high returns with less environment steps. Next, we focus on several meta-RL scenarios including ones with sub-tasks, varying dynamics and visual observations, and show that MAMBA outperforms all baselines in each. We also investigate properties of MAMBA compared to the baselines, such as robustness to hyperparameters and the ability to utilize information from previous sub-episodes. For visualizations of our results see https://sites.google.com/view/mamba-iclr2024, and for additional environments, experiments and details see Appendices C and H.

**Environments:** We use two common 2D environments in meta-RL, **Point Robot Navigation (PRN)**, and **Escape Room** (Zintgraf et al., 2019; Dorfman et al., 2021; Rakelly et al., 2019). In both, a goal is randomly chosen on a half-circle and the agent must explore the circumference until it reaches it. In **Point Robot Navigation** this point is a source of positive reward, and in **Escape Room**, it is a door that leads to high rewards located outside the circle. These represent basic scenarios for varying rewards and varying dynamics in meta-RL.

Next, to test our DTD hypothesis (Sec. 3.2), we design two scenarios composed of sub-tasks, one discrete (**Rooms-$N$**), and one continuous (**Reacher-$N$**) where $N$ represents the number of sub-tasks. In both tasks, a set of $N$ goals are selected independently, and the challenge is to reach them in the correct order and memorize them between sub-episodes. Crucially, the first $N-1$ goals only provide a positive reward once, thus encouraging the agent to find all $N$ goals and to stay on the last one as long as possible. In **Rooms-$N$** the minigrid agent starts in the leftmost room and the $i$-th goal is in the $i$-th room from the left (thus the last goal is on the rightmost room). In **Reacher-$N$**, the agent (adapted from the DeepMind Control Suite, Tunyasuvunakool et al. 2020) is a two-link robot and goals are placed in the reachable regions around it. Full details can be found in Appendix C. See Fig. 1 for visualizations of the scenarios and behaviors learned by MAMBA.

**Baselines and evaluation protocol:** To demonstrate the effectiveness of MAMBA, we compare it with the following baselines: two state-of-the-art meta-RL algorithms, **VariBad** and **HyperX**; an unmodified DreamerV3 which we dub **Dreamer-Vanilla**; and an otherwise unmodified DreamerV3 with hyperparameters identical to MAMBA, which we call **Dreamer-Tune**. Because we tried to make MAMBA as light as possible, the hyperparameter difference between Dreamer-Tune and Dreamer-Vanilla is in network sizes and policy imagination horizon (see Appendix D for details).

In all our experiments, we report the mean and standard deviation of four random seeds, as well as the number of environment steps each algorithm took to converge. In every experiment we test the best model seen during evaluation on a held-out test set of 1000 tasks.

**MAMBA is an effective meta-RL algorithm:** Our main results are described in Table 1. First, we observe that both VariBAD and HyperX, state-of-the-art algorithms specifically designed for meta-RL, perform worse than the model-based methods (both versions of Dreamer and MAMBA), both in final return and in convergence rate. This reinforces our observation regarding the similarity between both families of algorithms, and we attribute the performance gap to better algorithmic design choices in the Dreamer architecture (such as RSSM instead of simple RNN, and model-based policy optimization).

Next, we find that further fine-tuning Dreamer to meta-RL scenarios again boosts performance; focusing on Dreamer-Vanilla, Dreamer-Tune, and MAMBA, we see that it is inconclusive whether Dreamer-Vanilla and Dreamer-Tune is the better model. However, it is clearly evident that both are inferior to MAMBA. We note that the differences between Dreamer-Tune and MAMBA are algorithmic only (see Sec. 3.3), clearly demonstrating favorable algorithmic design choices in MAMBA.

Finally, we note that for the **sub-tasks scenarios** in particular, MAMBA achieves much better results than all baselines, in line with our hypothesis in Section 3.2.

**Robustness of MAMBA:** We evaluate robustness to hyperparameters on the Reacher-1 sceraio. Here, we take three VariBAD and HyperX hyperparameter configurations from the original papers and we compare them to both Dreamer and MAMBA. The hyperparameter configuration of Dreamer-Vanilla was taken from Hafner et al. (2023), which is almost identical to Dreamer-Tune

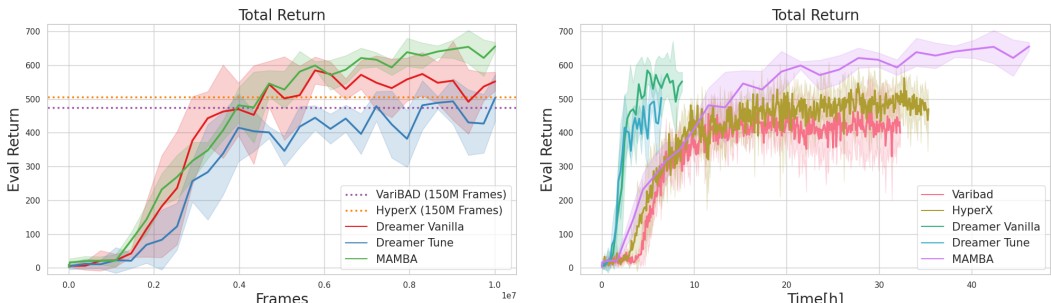

Figure 3: **Returns in Reacher-1:** Left: sample efficiency – meta-episode returns against the number of frames played. Since VariBAD and HyperX took 15× frames to converge we show their limit instead of a full plot. Right: time efficiency – meta-episode returns against training time (hours).

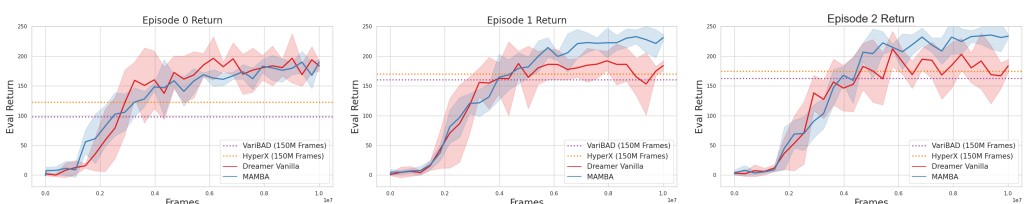

Figure 4: **Sub-episode Returns of MAMBA, VariBAD, HyperX and Dreamer-Vanilla in Reacher-1** (the meta-episode contains three sub episodes). Return of episode 0, 1, and 2, are left, middle, and right respectively. The gap between MAMBA and Dreamer demonstrates that MAMBA retains information from previous sub-episodes better than Dreamer.

and MAMBA (see Appendix D). Results for their respective configurations are in Table 1. As for VariBAD and HyperX, Table 1 contains the best configuration which was also used for the other Reacher tasks ($473.7 \pm 27.5$ for VariBAD, and $505.9 \pm 36.0$ for HyperX). The other two configurations of VariBAD achieved $104.6 \pm 34.3$ and $20.4 \pm 2.4$. In HyperX, one configuration resulted in unstable policies obtaining arbitrarily negative returns, and the last configuration obtained $14.3 \pm 0.4$. We conclude that these algorithms are highly sensitive to their hyperparameters.

This experiment hints that for new meta-RL scenarios, MAMBA and Dreamer provide good performance out-of-the-box without extensive hyperparameter tuning, while VariBAD and HyperX are more affected by hyperparameter tuning (see Figure 3, left).

**Is MAMBA better at exploitation than Dreamer?** In Sec. 3.3 we hypothesize that when allowing the RSSM to encode the full meta-episode, information regarding the MDP identity can be effectively retained between sub-episodes. In Fig. 4, we plot the return per sub-episode in the meta-episode. Returns for the first episode (left) are similar (hinting that both Dreamer and MAMBA learn to explore well), but MAMBA better utilizes the information gathered in previous episodes in the subsequent episodes reaching higher returns (middle and right).

**Visual domain:** In this scenario we leverage the ability of the Dreamer architecture to handle visual inputs. We consider two scenarios, a single goal Reacher scenario with two sub-episodes *with image observations*, and *Panda Reacher* (Choshen & Tamar, 2023), where a 7-DoF robot searches for goals with its end-effector. It is worth noting that VariBAD and HyperX do not support visual inputs. We ran each variant in *Reacher* and *Panda Reacher* for 15M and 20M environment steps respectively. Returns are $62.7 \pm 63.4$ and $96.3 \pm 26.6$ in Dreamer-Vanilla, $62.7 \pm 63.4$ and $111.1 \pm 28.8$ in Dreamer-Tune and $215.6 \pm 64.8$ and $137.6 \pm 0.7$ using MAMBA, correspondingly. Both scenarios show that MAMBA maintains superior returns in visual settings as well.

**Runtime limitation:** One limitation of our approach is runtime – to recall information from the start of the meta-episode, we unroll the RSSM over the entirety of the meta-episode. In some scenarios, this requires us to sequentially encode orders of magnitude more transitions.

| | VariBAD | HyperX | Dreamer-Vanilla | Dreamer-Tune | MAMBA (Ours) |
|---|---|---|---|---|---|
| PRN: | $218.0 \pm 26.0$ | $204.0 \pm 19.9$ | $\mathbf{241.1 \pm 9.9}$ | $\mathbf{239.7 \pm 3.1}$ | $\mathbf{242.2 \pm 7.9}$ |
| | $\sim$10M | $\sim$15M | $\sim$2M | $\sim$2M | $\sim$2M |
| Rooms-3: | $\mathbf{158.7 \pm 2.6}$ | $111.8 \pm 20.2$ | $137.5 \pm 6.2$ | $142.6 \pm 4.1$ | $\mathbf{156.2 \pm 1.7}$ |
| Rooms-4: | $88.1 \pm 39.2$ | $14.1 \pm 3.0$ | $115.0 \pm 6.7$ | $119.8 \pm 3.1$ | $\mathbf{136.7 \pm 1.4}$ |
| Rooms-5: | $0.9 \pm 2.5$ | $-15.8 \pm 0.4$ | $96.5 \pm 2.5$ | $93.9 \pm 5.3$ | $\mathbf{113.1 \pm 5.7}$ |
| Rooms-6: | $-11.0 \pm 0.9$ | $-15.7 \pm 0.7$ | $69.5 \pm 6.3$ | $71.0 \pm 6.3$ | $\mathbf{94.5 \pm 1.2}$ |
| | $\sim$100M | $\sim$100M | $\sim$6M | $\sim$6M | $\sim$6M |
| Reacher-1: | $473.7 \pm 27.5$ | $505.9 \pm 36.0$ | $552.0 \pm 27.8$ | $503.4 \pm 69.0$ | $\mathbf{655.5 \pm 12.3}$ |
| Reacher-2: | $46.6 \pm 33.2$ | $30.0 \pm 48.5$ | $\mathbf{247.4 \pm 80.5}$ | $\mathbf{217.6 \pm 64.3}$ | $\mathbf{285.8 \pm 89.6}$ |
| Reacher-3: | $0.2 \pm 0.2$ | $0.5 \pm 0.2$ | $183.6 \pm 100.0$ | $76.9 \pm 80.5$ | $\mathbf{325.0 \pm 47.0}$ |
| Reacher-4: | $0.0 \pm 0.0$ | $-0.5 \pm 1.1$ | $0.4 \pm 0.0$ | $0.1 \pm 0.2$ | $\mathbf{77.7 \pm 61.1}$ |
| | $\sim$150M | $\sim$150M | $\sim$10M | $\sim$10M | $\sim$10M |
| EscapeRoom: | $70.7 \pm 5.3$ | $66.9 \pm 6.5$ | $68.2 \pm 2.4$ | $\mathbf{73.2 \pm 7.8}$ | $\mathbf{73.9 \pm 3.1}$ |
| | $\sim$20M | $\sim$20M | $\sim$4M | $\sim$4M | $\sim$4M |

Table 1: Total return comparison of VariBad, HyperX, DreamerV3, and MAMBA on different meta environments. We report the final reward (mean $\pm$ std), and the number of time steps until convergence (below dashed line).

We investigate the runtimes of each of the baselines on the Reacher scenario with a single goal. Fig. 3 (right) shows the return of each algorithm compared to the runtime of the algorithm[1]. First, we note that both Dreamer-Tune and Dreamer-Vanilla are faster than MAMBA (Dreamer-Tune is slightly faster than Dreamer-Vanilla due to reduced NN sizes). However, MAMBA is on-par or better than both VariBAD and HyperX. The time gap between Dreamer and MAMBA could be further reduced, which is an interesting direction for future research. Another direction for future research is to "chunk" the meta-episode transitions in order to reduce the length of the RNN rollout. This requires further investigation, as it removes contextual information which may be required.

## 5 RELATED WORK

**Meta-Reinforcement Learning (meta-RL)** has been extensively studied in recent years (Duan et al., 2016; Wang et al., 2017; Finn et al., 2017). Of specific interest to our work are context-based meta-RL approaches (Rakelly et al., 2019; Zintgraf et al., 2019; 2021), which attempt to learn a Bayes-optimal policy, and are model-free, thus suffering from high sample complexity.

**Model-based RL** has been shown to be more sample efficient than its model-free counterparts (Nagabandi et al., 2018; Janner et al., 2019; Schrittwieser et al., 2020; Ye et al., 2021). While there are many different approaches to model learning, Dreamer is a prominent line of work (Hafner et al., 2019a;b; 2020; 2023; Mendonca et al., 2021), which learns a recurrent latent space world model and uses rollouts in latent space to train a policy. This approach has been shown to work well in POMDP environments, owing to the inherent RNN history aggregation ability (Ni et al., 2022).

**Model Based Meta-RL:** Recent approaches (Nagabandi et al., 2018; Perez et al., 2020; Lin et al., 2020; Hiraoka et al., 2021) all continuously adapt to a varied task distribution, making no distinction of when a new task is attempted. In accordance, they all use short histories of states and actions, limiting usability when long horizons are required for task identification. Rimon et al. (2022); Lee & Chung (2021) learn a model of the task distribution, sampling from it to generate tasks and using these to generate full trajectories. Conversely, in our work, imagined rollouts are history-dependent, allowing the model to incorporate information from online data. An extended related work section is available in Appendix F.

## 6 CONCLUSION

We present MAMBA, a high-performing and versatile meta-RL algorithm based on the Dreamer architecture. We show that MAMBA significantly outperforms existing meta-RL approaches in cumulative returns, sample efficiency and runtime. We find this result to be consistent in common meta-RL benchmarks, in tasks with visual inputs and in a class of decomposable task distributions, for which we also provide PAC bounds. In addition, MAMBA requires little to no hyperparameter tuning. Finally, we believe that MAMBA can be used as a stepping stone for future meta-RL research, generalizing to more challenging domains and real-world applications.

---

[1]All experiments were conducted using an Nvidia T4 GPU, with 32 CPU cores and 120GB RAM.

## REPRODUCIBILITY

Our implementation of MAMBA can be found in `https://github.com/zoharri/mamba` and is based on an open-source implementation of DreamerV3 (Hafner et al., 2023) found at `https://github.com/NM512/dreamerv3-torch`, borrowing elements from the open-source implementations of Zintgraf et al. (2019; 2021) at `https://github.com/lmzintgraf/hyperx`. All hyperparameters and details modified from the original implementation are described in the Experiments section (Sec. 4), as well as in Appendix D.

## 7 ACKNOWLEDGEMENTS

This work received funding from the European Union (ERC, Bayes-RL, Project Number 101041250). Views and opinions expressed are however those of the authors only and do not necessarily reflect those of the European Union or the European Research Council Executive Agency. Neither the European Union nor the granting authority can be held responsible for them.

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

## A  Appendix

In Section A.1 we define the term Decomposable Task Distribution and in Section A.2 we show (under some mild assumptions) that much more favorable bounds can be obtained for these distributions compared to the general case.

### A.1  Decomposable Task Distributions

We follow the notation from Rimon et al. (2022) and assume that a task distribution is defined by a parametric space $\Theta$, a prior distribution over this space $f \in \mathcal{P}(\Theta)$ and a mapping function from the parametric space to the MDP space $g : \Theta \to \mathcal{M}$. To sample an MDP from the distribution, a parameter is sampled from $f$ and is passed through $g$. Essentially, we assumes that the distribution over MDPs can be defined as a parametric variation. This formulation is very broad and can be used to formalize all the environments considered in this paper; we refer the reader to Rimon et al. (2022) for examples and motivation. Using this formulation we can write the expected return from Section 2.1 as:

$$J^\pi = \mathbb{E}_{\theta \sim f} \left[ \mathbb{E}_{\pi, M=g(\theta)} \left[ \sum_{k=1}^{H} r_{t,k} \right] \right]$$

We now formally define a *Decomposable Task Distribution*, which as its name suggests, is a task distribution where each task can be decomposed to subtasks.

**Definition 1** *Decomposable Task Distribution. Given* $\{t_i, f_i, \Theta_i, g_i\}_{i=1}^{N_T}$ *where* $t_i \in \mathbb{N}, \Theta_i \subset \mathbb{R}^{d_i}, d_i \in \mathbb{N}, f_i \in \mathcal{P}(\Theta_i), g_i : \Theta_i \to \mathcal{M}$ *and* $\mathcal{M}$ *is the MDPs space. To sample an MDP from this distribution the parameters are sampled from the priors and subtasks are being generated* $\theta_i \sim f_i$, $M_i = g_i(\theta_i), \forall i \in [1, N_T]$. *The resulting full MDP is defined by playing in* $M_i$ *between time steps* $t_i$ *and* $t_{i+1}$ $\forall i \in [1, N_T]$, *without resampling from the initial state distribution when passing from one MDP to another.*

### A.2  PAC Bounds Using Sub-Task Decomposition

**General Regret Bounds**  We first define the regret – the difference in the return of policy $\pi$ to the return of the Bayes-optimal policy – as follows:

$$\mathcal{R}(\pi) = J^{\pi_{BO}} - J^\pi = \mathbb{E}_{\theta \sim f} \left[ \mathbb{E}_{\pi_{BO}, M=g(\theta)} \left[ \sum_{t=1}^{H} r_t \right] - \mathbb{E}_{\pi, M=g(\theta)} \left[ \sum_{t=1}^{H} r_t \right] \right]$$

Where $\pi_{BO}$ is the Bayes-optimal policy, i.e $\pi_{BO} \in \arg\max_\pi J^\pi$. Note that the regret is 0 iff $\pi$ is Bayes-optimal. Since the regret is taken with respect to the unkown prior distribution, it is a measure of the generalization of $\pi$.

In their work, Rimon et al. (2022) suggest a density estimation approach to achieve regret bounds. Given a set of $N$ training samples, they first estimate the parametric distribution of tasks $f(\theta)$ with Gaussain Kernel Density Estimation (KDE, Rosenblatt 1956) $\widehat{f}_G(\theta)$. By using KDE error bounds (Jiang, 2017) they achieved probably almost correct (PAC) bounds on the regret:

**Theorem 2** *(Theorem 6 in Rimon et al. 2022) For a prior* $f(\theta)$ *over a bounded parametric space* $\Theta$ *that satisfies* $\|f\|_\infty < \infty$ *and is* $\alpha$-*Hölder continuous, and a Gaussian KDE, we have that with probability at least* $1 - 1/N$:

$$\mathcal{R}(\pi_{\widehat{f}_G}^*) \leq 2R_{max} H \, |\Theta| \, C_d \cdot \left( \frac{\log N}{N} \right)^{\frac{\alpha}{2\alpha+d}}.$$

Where $N$ are the number of sampled tasks, $R_{max}$ is the maximum possible reward, $|\Theta|$ and $d$ are the volume and dimension of the space $\Theta$, respectively, $C_d$ is polynomial of order $d$ and $\pi_{\widehat{f}_G}^* \in \arg\max_\pi J_{\widehat{f}_G}^\pi = \arg\max_\pi \mathbb{E}_{\theta \sim \widehat{f}_G} \left[ \mathbb{E}_{\pi, M=g(\theta)} \left[ \sum_{t=1}^{H} r_t \right] \right]$, i.e the Bayes-optimal policy under the assumption that the estimator $\widehat{f}_G$ is correct.

This theorem shows that, given a general parametric space, the number of task samples needed to guarantee a constant regret scales exponentially with the dimension of the parametric space. This observations shows an inherit limitation of meta-RL.

**Bounds for Decomposable Distributions** The obvious approach to achieve regret PAC bounds for decomposable task distributions, is to take a similar approach as done in Theorem 2. We can handle the decomposable distributions as normal distributions where $\Theta_{full} = \Theta_0 \times \Theta_1 \times \cdots \times \Theta_{N_T}$ and the prior distribution over the parametric space is $f^{full}(\theta = (\theta_1, \ldots, \theta_{N_T})) = \prod_{i=1}^{N_T} f_i(\theta_i)$. In that case we can approximate $f^{full}$ directly using a KDE $\widehat{f}_G^{full}$ over the samples $\{\theta_i\}_{i=1}^N$ and using Theorem 2 get regret bounds of:

$$\mathcal{R}(\pi^*_{\widehat{f}_G^{full}}) \leq 2R_{max}H\left|\Theta^{full}\right|C_{d^{full}} \cdot \left(\frac{\log n}{n}\right)^{\frac{\alpha}{2\alpha + d^{full}}} .$$

Where $d^{full} = \sum_{i=1}^{N_T} d_i$, with $d_i$ being the dimensionality of $\Theta_i$. This means that if we approach decomposable task distributions as regular distributions, to guarantee a constant regret we'll need exponentially growing number of samples with respect to $d_{full}$, i.e depend on both the number of sub-tasks and the number of DoF in each sub-task distribution.

We suggest a different approach: estimate each distribution $f_i$ by a KDE $\widehat{f}_i$ separately, i.e $\hat{f} = \{\hat{f}_i\}_{i=1}^{N_T}$. For a policy $\pi$ and a decomposable task distribution defined by $\{t_i, f_i, \Theta_i, g_i\}_{i=1}^{N_T}$, we define $\{g_i^\pi\}_{i=1}^{N_T}$, where $g_i^\pi(\theta) = g_i(\theta)$, except for the initial state distribution, which is set to be the induced initial distribution of the $i$'th MDP, as defined in Definition 1 $\forall \theta \in \Theta_i$ and $\forall i \in [1, N_T]$. Next we follow the derivation of Theorem 6 in Rimon et al. (2022):

$$J_f^\pi = \sum_{i=1}^{N_T} \mathbb{E}_{\theta \sim f_i}\left[\mathbb{E}_{\pi, M=g_i^\pi(\theta)}\left[\sum_{t=t_{i-1}}^{t_i} r_t\right]\right] = \sum_{i=1}^{N_T} \int \mathbb{E}_{\pi, M=g_i^\pi(\theta)}\left[\sum_{t=t_{i-1}}^{t_i} r_t\right] f_i(\theta)d\theta$$

$$J_f^\pi - J_{\widehat{f}_{Sub}}^\pi = \sum_{i=1}^{N_T} \int \mathbb{E}_{\pi, M=g_i^\pi(\theta)}\left[\sum_{t=t_{i-1}}^{t_i} r_t\right]\left(f_i(\theta) - \widehat{f}_i(\theta)\right) d\theta$$

And:

$$\left|J_f^\pi - J_{\widehat{f}_{Sub}}^\pi\right| \leq \sum_{i=1}^{N_T} \int \mathbb{E}_{\pi, M=g_i^\pi(\theta)}\left[\sum_{t=t_{i-1}}^{t_i} r_t\right]\left|f_i(\theta) - \widehat{f}_i(\theta)\right| d\theta$$

$$\leq \sum_{i=1}^{N_T} R_{max,i} \cdot (t_i - t_{i-1}) \int \left|f_i(\theta) - \widehat{f}_i(\theta)\right| d\theta$$

And following the same lines as the proof of Lemma 3 of Rimon et al. (2022) we get:

$$\mathcal{R}(\pi^*_{\widehat{f}_{Sub}}) \leq 2 \cdot \sum_{i=1}^{N_T} R_{max,i} \cdot (t_i - t_{i-1}) \int \left|f_i(\theta) - \widehat{f}_i(\theta)\right| d\theta$$

Using the PAC bounds for the KDE error of Jiang (2017), as was done in Rimon et al. (2022), we receive the final regret PAC bounds:

**Theorem 3** *For a decomposable prior distribution $\{t_i, f_i, \Theta_i, g_i\}_{i=1}^{N_T}$ where every parametric space is bounded, $\|f_i\|_\infty < \infty$ and $f_i$ is $\alpha$-Hölder continuous and for a set of Gaussian KDEs as defined above, we have that with probability at least $1 - 1/N$:*

$$\mathcal{R}(\pi^*_{\widehat{f}_{Sub}}) \leq 2R_{max}H|\Theta_{max}|C_{d_{max}} \cdot \left(\frac{\log N}{N}\right)^{\frac{\alpha}{2\alpha + d_{max}}} .$$

Where $N$ are the number of sampled tasks, $R_{max}$ is the maximum possible reward (over all the subtasks), $|\Theta_{max}|$ and $d_{max}$ are the maximal volume and dimensions of the spaces $\{\Theta_i\}_{i=1}^{N_T}$, respectively, $C_{d_{max}}$ is polynomial of order $d_{max}$ and $\pi^*_{\widehat{f}_{Sub}} \in \arg\max_\pi J^\pi_{\widehat{f}_{Sub}} = \mathbb{E}_{\theta \sim \widehat{f}_{Sub}} \left[ \mathbb{E}_{\pi, M=g(\theta)} \left[ \sum_{t=1}^H r_t \right] \right]$, i.e the Bayes-optimal policy under the assumption that the estimator $\widehat{f}_{Sub}$ is correct.

As we can see, using the fact that the distribution can be decomposed into sub-distributions and approximating each one individually, results in much more favourable bounds. With the new approach, to guarantee a constant regret we need exponentially growing number of samples with respect to $d_{max}$ instead of $d_{full}$. This essentially means that if we approach this kind of tasks by solving each sub-task individually, *the difficulty of the task will be determined by the sub-task with the highest number of DoF, and not by the number of sub-tasks.*

For example, looking at the multi-goal Rooms-$N$ environment from our experiments, although the total number of degrees of freedom is $N$, i.e grows as we increase the number of rooms, the difficulty of the task does not change, as the agent should solve each of the rooms separately. In comparison, our theorem states that a standard navigation goal in $N$ dimensions, which has the same number of DoF will be much harder since we must estimate all the DoF at once.

Therorem 3 also sheds light on our findings from Section 3.2 which show that VariBAD works better once the reconstruction window is smaller, which helps the belief (estimated by the latent vector) focus on the current sub-task.

## B    Differences Between VariBAD and Dreamer

For simplicity, we denote VariBAD as VB, and Dreamer as D.

- **History of optimization at timestep $t$:** In VB, when executing the optimization at time $t$, the entire meta-episode gets encoded into $b_t$. Conversely, in D, short ($L = 64$) sequences of transitions are sampled from the replay buffer and encoded into latent vectors $z_{t:t+L}$ using the RSSM world model. This is problematic in the meta-RL context, since it does not allow components (policy, model, value) to depend on the full history of the meta-episode and may lead to ambiguity that should have already been resolved.

- **World model prediction horizon:** In D, the world model predicts a single step, but forces the latent states of subsequent transitions to be consistent. In VB, the decoder directly predicts outcomes (next states and rewards) for every possible observation by feeding the belief, current observation and action as inputs to the decoder [2].

- **Horizon differences:** In VB, episodes were short (about 100 time steps), while in D, POMDPs with hundreds of time steps were considered. This technical consideration is probably one of the factors why D samples sub-trajectories of length $L$, and why VB were able to run on a GPU without out-of-memory exceptions. However, it is a limiting factor for many meta-RL tasks that are characterized with a long horizon due to the combination of several sub-episodes in every meta-episode.

- **Policy optimization:** In VB an online model-free algorithm PPO (Schulman et al., 2017) was used while D use a combination of the REINFORCE (Sutton et al., 1999) algorithm and backpropagation through trajectories imagined by the world model.

- **Policy input:** In VB, the policy is conditioned on both the latent $z_t$ and the current state $s_t$. However, in D the policy is conditioned only on $z_t$.

- **Hidden sizes in RNN:** D has much larger hidden layer sizes than VB.

## C    Full Environments Technical Details

Here we provide the full details regarding the environments we used in our experiments (Section 4).

---

[2]In VariBAD, observations were not used, so an observation in this context is equal to state.

**Point Robot Navigation**    A commonly used 2D continuous control benchmark for meta-RL (Zintgraf et al., 2019; Rimon et al., 2022; Dorfman et al., 2021; Choshen & Tamar, 2023), where a goal is sampled on a semi-circle with radius of 1.0. The reward is sparse and defined as $1-\|p_{agent}-p_{goal}\|_2$ when the agent is inside the goal radius (0.2) and 0 otherwise[3] (where $p_{agent}$, $p_{goal}$ are the agent and goal positions, respectively). Here, a meta-episode consists of two sub-episodes of $K = 100$ steps each. The Bayes-optimal behavior is to explore the semicircle on the first episode and exploit the information acquired by going directly to the goal on the second.

**Point Robot Navigation - Wind**    This is an extension of the Point Robot Navigation environment, where a stochastic wind force is added at each step. More explicitly, at each step, a 2D Gaussian random vector with controllable std (e.g. 0.3) is sampled and added to the selected action. Other than that, the environment details and Bayes-optimal behavior are the same as the in the Point Robot Navigation environment.

**Escape Room**    In this environment, taken from Dorfman et al. (2021), a 2D point robot is initialized in a circular room of radius 1.0. An arched door of length $\frac{\pi}{8}$ is randomly sampled on the upper half of the room, providing the only exit from the room. The reward is 0 inside the room and 1 outside of it. Here, a meta-episode consists of two sub-episodes of $K = 60$ steps each. A Bayes-optimal agent should explore the upper half circle, pushing against the wall until successfully exiting.

**Humanoid-Dir**    First proposed in Rakelly et al. (2019), in this environment the Humanoid MuJoCo (a high dimensional humanoid agent) task is initialized with a random target walking direction $\theta \in [0, 2\pi]$ on the 2D plane. The agent receives a reward proportional to the projection of its speed on the unknown goal direction. Since in meta-RL we observe the environment rewards, the Bayes-optimal behavior should be able to detect the correct direction after a few steps (theoretically one step) and consequently walk as fast as possible towards this direction.

**Reacher**    This environment, available in the DeepMind Control Suit (DMC, Tunyasuvunakool et al. 2020), is a planar two-link robot tasked to reach various goals on the plane with the tip of its second link. A goal is sampled on the 2D plane within a ball around the agent, which when reached (to a certain small distance), provides a reward proportional to the distance from the goal. For this environment we use 3 sub-episodes of length $K = 300$ in every meta-episode. In contrast to the point robot navigation task, since the goal can be sampled anywhere within the circle, the Bayes-optimal policy should explore in a spiral form to cover the entire goal space.

We also consider a multi-goal variant of this task, **Reacher-$N$**, with observation and action spaces identical to the original Reacher task. Here, $N$ goals are sampled i.i.d. (according to the distribution above) which the robot must reach in a predetermined order. The reward for each goal $[1, \ldots, N]$ may be obtained only after all the previous goals were reached. For the first $N - 1$ goals, the agent receives positive reward *only once* when the agent first penetrates the goal radius. In the $N$-th goal, the positive reward is given every step, until the sub-episode terminates. The Bayes-optimal behavior in this case is to explore until the last goal is found, and then stop moving. Once a goal was identified, in subsequent sub-episodes the policy should head to that identified goal directly. This serves as a high-dimensional environment decomposable into independent sub-tasks (up to the initial distribution), matching our analysis in Sec. 3.2. We note that for 3 and 4 goals we increased the number of steps per sub-episode to 400 instead of 300.

**Panda Reacher**    Proposed by Choshen & Tamar (2023), Panda Reacher is a dynamically complex system with visual inputs and sparse rewards which requires a non-trivial Bayes-optimal behavior. In this environment, a 7-degree-of-freedom (DoF) robotic arm is required to reach goals with the robot end-effector (i.e. the tip of the robot). The goals are hidden and must be discovered through

---

[3]Inside the goal radius the rewards are positive.

| Hyper-parameter | Dreamer-Vanilla | Dreamer-Tune | Description |
|---|---|---|---|
| dyn_discrete | 32 | 16 | Number of discrete latent vectors in the world model. |
| dyn_hidden | 512 | 64 | MLP size in the world model. |
| dyn_stoch | 32 | 16 | Size of each latent vector in the world model. |
| imag_horizon | 15 | 10 | Length of the imagination horizon ($N_{IMG}$ cf. Sec.2.2). |
| units | 512 | 128 | Size of hidden MLP layers. |

Table 2: Hyper parameters differences between Dreamer-Vanilla and Dreamer-Tune.

|  | Rooms-4 | Rooms-5 |
|---|---|---|
| **VariBAD Vanilla** | 14.1 +- 3.0 | 0.9+-2.5 |
| **VariBAD Local** | 88.1+-39.2 | 15.1+-13.1 |
| **MAMBA** | 136.7±1.4 | 113.1±5.7 |

Table 3: VariBAD reconstruction ablation study. Using local reconstruction helped in the Rooms-$N$ environment, which contains sub-tasks.

exploration. The rewards are sparse, meaning only when the end-effector is within a short distance from the goal, a reward of 1 is obtained (otherwise the reward is 0). The observation is an image of size $84 \times 84 \times 4$, and the action space is the direction towards which the end-effector is moved.

**Rooms-$N$:** As an additional multi-goal task, we consider a discrete grid-world environment consisting of a sequence of $N$ rooms of size $3 \times 3$ separated by a $1 \times 3$ corridor. At each location the agent observes its $(x, y)$ location and can choose to go right/left/up/down (will stay in place if the resulting state is out of the map). A goal is sampled in the corner of each room and the agent starts in the left-most room. The goals in rooms $[2, \ldots, N]$ are obtainable if all the previous goals were visited. The agent receives a reward of 1 when a goal is visited, and $-0.1$ otherwise. For the first $N - 1$ rooms, once a goal is visited in a sub-episode, it no longer emits a positive reward. On the first sub-episode, a Bayes-optimal policy should explore each of the rooms, continuing to the next room upon finding the goal. In the second sub-episode, it should go directly to the goal in each room. This environment serves as an additional sub-task decomposable environment (see Sec. 3.2).

## D   Hyperparameters of Dreamer-Tune

Our parameters of Dreamer-Vanilla are identical to DreamerV3 (Hafner et al., 2023). In Table 2 we list the differences between Dreamer-Vanilla and Dreamer-Tune.

## E   Empirical Results for VariBAD with Local Reconstruction

In Sec. 3.2, we introduced a variant of VariBAD that uses a local reconstruction (reconstructing a window of adjacent transitions instead of the full trajectory). In this section we evaluate it against the original VariBAD on two sub-task decomposable environments: Rooms-4 and Rooms-5. Results can be found in Table 3. We also provide the results of MAMBA from Sec. 4 as reference.

As we can see in Table 3 local reconstruction helped VariBAD in these scenarios, which is in line with our theoretical findings (cf. Sec. 3.2).

## F   Extended Related Work

**Meta-Reinforcement Learning (meta-RL)** has been extensively studied in recent years (Duan et al., 2016; Wang et al., 2017; Finn et al., 2017; Beck et al., 2023). Of specific interest to our work are context-based meta-RL approaches (Rakelly et al., 2019; Humplik et al., 2019; Zintgraf et al., 2019; 2021; Wang & Van Hoof, 2022), which attempt to learn a Bayes-optimal policy. Most of these are model-free, thus suffering from high sample complexity.

Others, such as Humplik et al. (2019), utilize access to privileged information during the meta-training phase and thus do not apply to the general setting of meta-RL.

**Model-based RL** has been shown to be more sample efficient than its model-free counterparts (Nagabandi et al., 2018; Clavera et al., 2018; Janner et al., 2019; Schrittwieser et al., 2020; Ye et al., 2021). While there are many different approaches to model learning and usage, a prominent line of work is the Dreamer framework (Hafner et al., 2019a;b; 2020; 2023; Mendonca et al., 2021), which learns a recurrent latent space world model and uses rollouts in latent space to train a policy. This approach has been shown to work well in POMDP environments, owing to the inherent history aggregation ability of the RNN (Ni et al., 2022).

**Model Based Meta-RL:** Recent approaches (Nagabandi et al., 2018; Perez et al., 2020; Lin et al., 2020; Lee et al., 2020; Hiraoka et al., 2021; Wang & Van Hoof, 2022; Pinon et al., 2022) propose model-based meta-RL methods. Nagabandi et al. (2018) use a dynamics model and adapt it online to varying dynamics in robotic scenarios. Perez et al. (2020) propose to encode the parameters controlling the reward and transition functions into latent variables of a learned model. To tackle the problem of task distribution shift, Lin et al. (2020) train a shared dynamics model on a distribution of tasks, but train a policy at test-time given a new reward function, preventing fast adaptation to new tasks. Lee et al. (2020) trains a context vector to be consistent with forward and backward dynamics, conditioning on this context to enable a dynamics model to generalize quickly. The above approaches all continuously adapt to a varied task distribution, making no distinction of when a new task instance is attempted. In accordance, they all use short histories of states and actions, limiting usability when long horizons are required for task identification.

An approach that does focus on long horizons and is designed specifically to test model memory is a baseline proposed in Pasukonis et al. (2022), where a variation of Dreamer-v2 (Hafner et al., 2020) is used to benchmark a new environment. While this baseline bears some similarities to our approach, Pasukonis et al. (2022) aims to establish a baseline for testing memory capabilities of agents, rather than proposing an algorithmic solution to meta-RL in general. In accordance, the modifications they propose for Dreamer are only geared at extending its memory capabilities, disregarding other modifications we found important in this work.

Wang & Van Hoof (2022) propose a GNN-based model learned for each task of the training distribution, along with an amortized policy selection method to quickly find the correct model at meta-test time. In contrast, our method learns a single model and a single policy for the entire task distribution. Pinon et al. (2022) propose a Transformer-based world model, using tree search to plan with the model at test time. While general, this approach may be limited to discrete-action domains; in addition, tree search is a time consuming process. Our method, on the other hand, can operate on discrete and continuous domains (as it is based on the Dreamer architecture), and learns a policy so that it can act quickly at test time without additional planning.

Rimon et al. (2022) and Lee & Chung (2021) learn a model of the task distribution, sampling from it to generate tasks and using these to generate full trajectories. Conversely, in our work, imagined rollouts are history-dependent, allowing the model to incorporate information from online data.

## G   ABLATION STUDY OF MAMBA

In this appendix we ablate the three different modifications proposed in Section 3.3 to identify the contribution of each.

First, we note that it is infeasible to run experiments without adding the reward into the observation as it prevents any possibility of the agent distinguishing between tasks; to validate this, we removed the reward observation and tested on the Rooms-3 environment. Results are as expected: compared to MAMBA, which obtained a return of $156.2\pm 1.7$, MAMBA w/o reward in the observation obtained only $11.5\pm 1.4$.

Next, MAMBA with the original batch sampling (i.e. batches do not start from the start of the meta-episode) is in fact Dreamer-Tune; therefore, we did not run a separate ablation test for this design choice.

Finally, we tested the effects of removing the horizon scheduling in the Point Robot Navigation scenario, and indeed found that the results obtained are on-par but runtime is slower (on the same hardware), see Table 4.

|  | MAMBA | MAMBA w/o Horizon Scheduling |
|---|---|---|
| Average Total Return: | $156.2 \pm 1.7$ | $\mathbf{156.8 \pm 1.2}$ |
| Total Time [H]: | 34.1 | 44.0 |

Table 4: Ablating the horizon schedule in the Point Robot Navigation environment.

|  | $RL^2$ | VariBAD | HyperX | Dreamer-Vanilla | Dreamer-Tune | MAMBA (Ours) |
|---|---|---|---|---|---|---|
| PRN: | $239.4 \pm 3.1$ | $218.0 \pm 26.0$ | $204.0 \pm 19.9$ | $\mathbf{241.1 \pm 9.9}$ | $\mathbf{239.7 \pm 3.1}$ | $\mathbf{242.2 \pm 7.9}$ |
|  | $\sim$10M | $\sim$10M | $\sim$15M | $\sim$2M | $\sim$2M | $\sim$2M |
| Rooms-3: | $108.0 \pm 31.7$ | $\mathbf{158.7 \pm 2.6}$ | $111.8 \pm 20.2$ | $137.5 \pm 6.2$ | $142.6 \pm 4.1$ | $\mathbf{156.2 \pm 1.7}$ |
| Rooms-4: | $85.1 \pm 19.0$ | $88.1 \pm 39.2$ | $14.1 \pm 3.0$ | $115.0 \pm 6.7$ | $119.8 \pm 3.1$ | $\mathbf{136.7 \pm 1.4}$ |
| Rooms-5: | $72.4 \pm 36.8$ | $0.9 \pm 2.5$ | $-15.8 \pm 0.4$ | $96.5 \pm 2.5$ | $93.9 \pm 5.3$ | $\mathbf{113.1 \pm 5.7}$ |
| Rooms-6: | $64.9 \pm 15.9$ | $-11.0 \pm 0.9$ | $-15.7 \pm 0.7$ | $69.5 \pm 6.3$ | $71.0 \pm 6.3$ | $\mathbf{94.5 \pm 1.2}$ |
| Rooms-7: | $42.9 \pm 22.9$ | - | - | - | $50.2 \pm 4.4$ | $\mathbf{73.2 \pm 2.9}$ |
| Rooms-8: | $29.0 \pm 17.0$ | - | - | - | $29.1 \pm 8.3$ | $\mathbf{55.6 \pm 3.0}$ |
|  | $\sim$100M | $\sim$100M | $\sim$100M | $\sim$6M | $\sim$6M | $\sim$6M |
| Reacher-1: | $595.8 \pm 21.4$ | $473.7 \pm 27.5$ | $505.9 \pm 36.0$ | $552.0 \pm 27.8$ | $503.4 \pm 69.0$ | $\mathbf{655.5 \pm 12.3}$ |
| Reacher-2: | $\mathbf{404.0 \pm 35.9}$ | $46.6 \pm 33.2$ | $30.0 \pm 48.5$ | $247.4 \pm 80.5$ | $217.6 \pm 64.3$ | $285.8 \pm 89.6$ |
| Reacher-3: | $\mathbf{390.6 \pm 49.1}$ | $0.2 \pm 0.2$ | $0.5 \pm 0.2$ | $183.6 \pm 100.0$ | $76.9 \pm 80.5$ | $325.0 \pm 47.0$ |
| Reacher-4: | $0.0 \pm 0.0$ | $0.0 \pm 0.0$ | $-0.5 \pm 1.1$ | $0.4 \pm 0.0$ | $0.1 \pm 0.2$ | $\mathbf{77.7 \pm 61.1}$ |
|  | - | $\sim$150M | $\sim$150M | $\sim$10M | $\sim$10M | $\sim$10M |
| EscapeRoom: | $\mathbf{79.9 \pm 4.4}$ | $70.7 \pm 5.3$ | $66.9 \pm 6.5$ | $68.2 \pm 2.4$ | $73.2 \pm 7.8$ | $73.9 \pm 3.1$ |
|  | $\sim$20M | $\sim$20M | $\sim$20M | $\sim$4M | $\sim$4M | $\sim$4M |

Table 5: Total return comparison of $RL^2$, VariBad, HyperX, Dreamer-Vanilla, Dreamer-Tune, and MAMBA on different meta environments. We report the final reward (mean $\pm$ std), and the number of time steps until convergence (below dashed line).

# H ADDITIONAL EXPERIMENTS

## H.1 ADDITIONAL META-RL BASELINES

### H.1.1 $RL^2$

$RL^2$ (Duan et al., 2016) is a common meta-RL baseline; our additional experiments (Table 5) show that MAMBA is superior to $RL^2$ in all environments except EscapeRoom, where the difference is slightly in favor of $RL^2$ (although $RL^2$ used $5\times$ more data than MAMBA).

### H.1.2 DREAM

DREAM (Liu et al., 2021) is a meta-RL algorithm with a more sophisticated out-of-the-box exploration mechanism. Since the experiments in DREAM deal with goal conditioned meta-RL tasks in grid-world environments, we tested it in our Rooms environment.

Our experiments in Rooms-3 show that DREAM fails to obtain any meaningful policy, and the meta-episode rewards remain negative (MAMBA, Dreamer, VariBAD and HyperX all managed to obtain returns greater than 100). Results from experiments with more rooms behave similarly. This may result from bad hyper-parameter adaptation between the scenarios in the DREAM paper and Rooms; however, this is another advantage of using a Dreamer-based architecture as in MAMBA (same hyperparameters fit different scenarios out-of-the-box).

## H.2 ADDITIONAL ENVIRONMENTS

### H.2.1 ROOMS-7 AND ROOMS-8

We added more rooms to challenge MAMBA. We observe that MAMBA still manages to find a Bayes-optimal behavior and the drop in return is attributed to the exploration required to find the additional goals. VariBAD or HyperX were not tested as they already failed to solve scenarios with less rooms. See Table 5.

|  | VariBAD | HyperX | Dreamer-Vanilla | Dreamer-Tune | MAMBA |
|---|---|---|---|---|---|
| Return: | $1369.3 \pm 75.3$ | - | $2068.3 \pm 156.7$ | $2096.3 \pm 79.8$ | **$2405.9 \pm 119.0$** |
| Steps: | 100M | - | 30M | 30M | 30M |

Table 6: Comparison of VariBad, HyperX, Dreamer-Vanilla, Dreamer-Tune, and MAMBA in the Humanoid-Dir environment.

|  | VariBAD | HyperX | Dreamer-Vanilla | Dreamer-Tune | MAMBA |
|---|---|---|---|---|---|
| Return: | $194.5 \pm 44.7$ | $177 \pm 118.5$ | $226.1 \pm 3.5$ | $224.5 \pm 4.4$ | $224.1 \pm 5.2$ |
| Steps: | 20M | 20M | 2M | 2M | 2M |

Table 7: Comparison of VariBad, HyperX, Dreamer-Vanilla, Dreamer-Tune, and MAMBA on the Point Robot Navigation - Wind environment.

### H.2.2 HUMANOID-DIR

First proposed in Rakelly et al. (2019), in this environment the Humanoid MuJoCo task is initialized with a random target walking direction $\theta \in [0, 2\pi]$ on the 2D plane. Table 6 summarizes the results of this experiment. As in the results described in Sec. 4, using less environment steps (30M vs. 100M) the algorithms based on the Dreamer architecture perform better than VariBAD and HyperX. Among these, MAMBA is substantially better, reaching the highest return.

### H.2.3 POINT ROBOT NAVIGATION - WIND

This is a modification of the Point Robot Navigation (PRN) environment, where a stochastic wind force is added at each step. Results in Table 7 show that the conclusions from Point Robot Navigation also extend to this stochastic version.

