# OpenReview forum: "MAMBA: an Effective World Model Approach for Meta-Reinforcement Learning"
_ICLR.cc/2024/Conference — ICLR 2024 poster_

### Official Review · Reviewer_tjN7 · 2023-10-22

**Soundness:** 3 good
**Presentation:** 3 good
**Contribution:** 3 good
**Rating:** 6
**Confidence:** 4

**Summary:**

This paper proposes a model-based method for Meta-RL. The method is based on the Deamer-v3 algorithm, which is already good at solving POMDPs. They propose three changes to Dreamer-v3, that make it better suited to meta-RL style POMDPs specifically. The changes are training the model on full episodes, scheduling the training horizon length, and adding rewards and timesteps to the observations. They test the proposed model on meta-RL tasks and show that it works well. Additionally, a theoretical result is presented, which shows that decomposable task distributions can enable much lower sub-optimality than non-decomposable task distributions. Using that theoretical result they suggest a simple empirical modification applicable to VariBAD, which achieves better results in the decomposable task distributions. The proposed method naturally enjoys the same benefit.

**Strengths:**

## Contribution
- Dreamer-style model-based RL algorithms seem like a natural fit for meta-RL problems. This paper shows that with few simple modifications, Dreamer-v3 performs well in meta-RL POMDPs.
- The theoretical result about decomposable task distributions is interesting and nicely supported by the experiments.
- The empirical results support the design choices and the theoretical results

## Presentation
- The paper is well written for the most part.

**Weaknesses:**

## Benchmark selection
- The environments used in the empirical section nicely illustrate the theoretical points and the advantage from the design choices, but they still feel a little limited considering the POMDPs dreamer-v3 is capable of solving. I would not want you to drop any of the domains already included, but it would make the paper stronger to run the algorithms on something more complex than reacher, which is from the easier end of mujoco tasks.
- It would be good to run the algorithm on at least some common meta-RL environments to help people who are familiar with that literature to ground the performance. I have no doubt that this works well there but it would be reaffirming to see those results. Consider running this for example in Walker and Humanoid tasks.
- In the theory section, it is discussed that the local reconstruction could be harmful in high dimensional non-decomposable task distributions. It would have been good to include a demonstration of such failure mode in the experimental section.

## Presentation
- I found it confusing that the results from the room experiments were discussed in 3.2.2. before the room experiments were described.
- Figure 2 is very small and busy. Neither 3.2.2 nor the caption contains enough detail to understand what is going on in the figure.
- The sentence in parenthesis in the first paragraph of 3.2.2 is hard to interpret. Maybe a typo?

**Questions:**

- The proposed method seems quite close to [1]. It would be good to add this in the related work section and discuss how they differ.
- What are some examples of task distributions where the task dimension is high?

[1] Pasukonis et al., 2022, Evaluating Long-Term Memory in 3D Mazes

---

> ### Author Response · Authors · 2023-11-21
> **Response to Reviewer tjN7**
>
> We thank you for your time and constructive review.
>
> 1. **Regarding harder environments**, we first clarify why the Reacher environment is complex and then describe three additional experiments (found in the new additional experiments supplementary material)
>    * We emphasize that we chose Reacher because we wanted to demonstrate that Dreamer style architectures are able to handle tasks where the difficulty is attributed to the meta learning aspect (variation of the dynamics and rewards) and not necessarily the dynamics themselves because previous works showed that Dreamer is able to solve tough control problems effectively. Besides the inherent difficulty in Reacher1, when adding multiple goals the problem becomes much harder as can be seen by the difficulty of VariBAD \ HyperX to adapt to those environments.
>    * We added two challenging environments, Humanoid-Dir and PandaReach. In both we see that MAMBA and the Dreamer versions are superior to VariBAD and HyperX both in return and in required environment steps to reach conversion. For full details about these new scenarios see the additional experiments supplementary file we added.
>    * Finally, we also added additional experiments for a harder version of Rooms, with seven and eight rooms, for the purpose of discovering Mamba’s performance limit in this domain. We find that even here only Mamba is able to find Bayes optimal behaviors and the drop in return is due to the fact that more exploration steps are required for the additional rooms.
>
> 2. We also considered **more common meta-RL benchmarks** such as HalfCircle and Humanoid-Dir. Another important aspect is the sparsity of rewards – many meta-RL environments are dense, i.e. the task could be inferred from all \ many actions available to the agent, making the Bayes-optimal behavior trivial! Many of the scenarios we investigated (Rooms, Reacher) are sparse (i.e. sub-goals can only be inferred when encountered) thus require non-trivial exploration policies. Nonetheless, we conduct additional experiments on Humanoid (see above) as we appreciate the need for standard benchmarking.
>
> 3. **Regarding “In the theory section, it is discussed that the local reconstruction could be harmful in high dimensional non-decomposable task distributions”**:
> Previous works [2],[3] have shown that scenarios of high-dimensional task distributions are hard for every meta-RL algorithm. Therefore, it is not our method specifically that is limited, but every meta-RL approach. Despite that, some methods might be more tailored towards specific aspects of the meta-RL problem. In our paper we demonstrated that our algorithm handles decomposable tasks better than previous approaches, even though they are high dimensional, but we also showed that it is better in more common scenarios (e.g. Rooms-1 and Reacher-1).
>
> 4. Some examples of **high dimensional task distributions** are the decomposable tasks in our work; while each sub-task in itself is of lower dimension, solving a sequence of them is high-dimensional. Other examples are domains such as Meta-World or RLBench; these are hard for all meta-RL approaches, and the current SOTA for these domains are based on fine tuning, which requires orders of magnitude more samples at test time than the meta RL approach we consider.
>
> 5. We incorporated suggestions on presentation and proposed related work into the relevant sections in the paper. Please see the revised version (revisions are in blue text).
> We acknowledge that the specific paper mentioned by the reviewer, Pasukonis et al. [1], bears some similarities to our approach: the Dreamer (TBTT) baseline in the paper is a modification to Dreamer-v2 designed to consider long-term memory effects, and the MemoryMaze environment is somewhat reminiscent of our Rooms-N environment. However, Pasukonis et al. [1] aims to establish a baseline for testing memory capabilities of agents, rather than proposing an algorithmic solution to meta-RL in general. Therefore, the modifications they propose for Dreamer are only geared at extending its memory capabilities, disregarding other modifications we found important.
>
> [1] Pasukonis et al., 2022, Evaluating long-term memory in 3D mazes
>
> [2] Mandi et al., 2022, On the effectiveness of fine-tuning versus meta-reinforcement learning
>
> [3] Rimon et al., 2022, Meta reinforcement learning with finite training tasks-a density estimation approach.

---

> > ### Comment · Reviewer_tjN7 · 2023-11-23
> >
> > Thanks for the thorough response. It addressed most of my original concerns. I have read the other reviews and believe they raise enough good questions about the quality of the current version, that raising my score to 8 is not justified. I know that responding to a wide range of questions in a tight rebuttal period, especially when reviewers ask for new experiments, is really challenging. But the additions are somewhat lacking in their design and presentation. Therefore, the total effect is that the paper does not seem significantly improved by the new edits.
> >
> > Comments based on the other reviews and the newest revision
> > - Why are the humanoid results are not in the appendix? In general, I don't understand the difference between the supplementary file and the appendix.
> > - Why does the no-horizon scheduling ablation study in the appendix have vastly different results than the one reported in the response to fKkT? The results in the appendix show that the MAMBA works much better w/o horizon scheduling. (Average Total Return: w/ horizon scheduling: 1369.3 ± 75.3, w/o horizon scheduling: 2405.9 ± 119.0). Furthermore the Table 4 where the horizon scheduling results are reported is not referred to anywhere.
> > - The figures in the newest revision are not as clear as they could be
> >     - Figure 2 has very small and hard to interpret subfigures
> >     - The coloring of the curves between the panels in figure 3 is not consistent.
> > - The addition of the DREAM baseline suggested by fKkT does not build confidence in the empirical evaluation as a whole.

---

> > > ### Author Response · Authors · 2023-11-23
> > > **Additional clarification**
> > >
> > > Thanks again for taking time to address our rebuttal.
> > >
> > > Regarding the Humanoid experiment (and the supplementary PDF in general), as well as the styling suggestions: we will include all results in the final version of the paper, and attempt to improve the visual quality of the figures. In addition, we can extend the DREAM comparison to make a more compelling case; note, however, that we used DREAM as intended (using the “official” implementation and no hyper-parameter tuning), as mentioned in the final reply to reviewer fKkT.
> > >
> > > Regarding the horizon scheduling ablation: we apologize for the confusion, the results in the appendix were reported in error. The correct results are the ones we attached to the reply to reviewer fKkT. We will fix the appendix accordingly for the final version of the paper.

---

### Official Review · Reviewer_X5No · 2023-10-29

**Soundness:** 2 fair
**Presentation:** 3 good
**Contribution:** 2 fair
**Rating:** 6
**Confidence:** 4

**Summary:**

In this paper, the authors tackle the problem of meta-RL: learning across a range of tasks at the same time. They combine the sample efficiency of model-based RL with an existing meta-RL framework to produce a method (Mamba) which is more sample efficient and powerful than existing methods on a range of meta-RL tasks including a set of novel tasks which are benchmarked against other methods.

The authors provide theoretical results comparing VariBAD (a meta-RL algorithm) with Dreamer (a model-based algorithm) on decomposable tasks which gives grounds for why such tasks are hard for model-free meta-RL algorithms. They then provide experimental results over a number of meta-RL tasks providing evidence for the improvements that Mamba has over existing methods.

The appendix goes into more detail on the theoretical and technical details.

**Strengths:**

Based on the evidence provided, it is clear that within the domains examined, Mamba is a stronger meta RL-algorithm than those tested against both in terms of returns and sample efficiency. The results are clearly presented and the explanations are generally sufficient.

**Weaknesses:**

While overall the paper is good, and provides good evidence within the context, it seems that there are a number of approaches within the literature which have not been covered. These include:

1) Pinon et al, A model-based approach to meta-reinforcement learning:transformers and tree search, https://arxiv.org/pdf/2208.11535.pdf
2) Wang and Hoof, Model-based meta reinforcement learning using graph structured surrogate models and amortized policy search, https://proceedings.mlr.press/v162/wang22z/wang22z.pdf
3) Clavera et al, Model-Based Reinforcement Learning via Meta-Policy Optimization, https://arxiv.org/pdf/1809.05214v1.pdf
4) Lee et al, Context-aware Dynamics Model for Generalization in Model-Based Reinforcement Learning, https://arxiv.org/abs/2005.06800

While there is a short section with a few papers on model-based RL, it seems that there are some important pieces of research which have been missed out here. Given how much has been missed, it is not clear that the results are as persuasive as they at first seem.

In addition, there exists an extensive meta-RL benchmarking suite produced by Wang et al, Alchemy: A benchmark and analysis toolkit for meta-reinforcement learning agents, https://arxiv.org/abs/2102.02926. I believe that it is vital for the community to come together to standardise benchmarking, and so such a toolkit seems ideal to truly show the applicability and strengths of Mamba.

On a stylistic note, there are many typos throughout, particularly in the Appendix. These, along with points of clarity are explained here:

1) Section 3.2: family task-> family of task
2) Section 3.2.1 PAC not defined
3) S 3.2.1: with high number->with a high number
4) S 3.3: singnals->signals
5) The plot in figure 2 is not clear given that the environment has not been clearly explained by this point.
6) Sometimes Dreamer-tune is written and sometimes Dreamer-tuned
7) In section 4, figure 1 is mentioned, which appears very close to the beginning of the paper, without context and is unclear at this point. It should be later in the paper when it is introduced.
8) Figure 4: amd->and
9) Table 1: Bold font on the 7 in 73.9±3.1
10) As discussed above, I believe that a lot of model-based meta-RL work has been left-out.
11) A1 folmulation->formulation
12) Definition 1 \forall 1<=i<=N_T should be written \forall i \in [1,N_T]
13) A2: well known Regret->Regret
14) Bayes optimal does not have consistent hyphenation or capitalisation
15) Above theorem 2: boudns->bounds
16) Theorem 2: polynomial with d->polynomial of order d
17) assumption the estimator->assumption that the estimator
18) Top of page 14: bounds in the private case-> ?
19) First equation on page 14: C_d should be C_{d^full}
20) guerntee-> guarantee
21) Proof of Lema 3->Proof of Lemma 3
22) Theorem 3: distriubtion->distribution
23) Theorem 3: Something is wrong after alpha-Holder continuous
24) polynomial with d_max should again be "of order"
25) assumption the estimator->assumption that the estimator
26) using the fact the->using the fact that
27) much favourable->much more favourable
28) this kind of tasks->these kinds of tasks
29) same amount of DoF->same number of DoF
30) we have estimate->we have estimated
31) Appendix B: short L=64->short (L=64)
32) Check consistency of hyphenation in world-model throughout
33) short 100 timesteps -> short (L=100)
34) Top of page 16: p_agent and p_goal not defined
35) Appendix E: It's not clear if there should be more in this section.

As can be seen, there are a lot of typos in the appendix which have taken away time from more important aspects of the paper.

**Questions:**

The questions are all based on the weaknesses in the previous section.

---

> ### Author Response · Authors · 2023-11-21
> **Response to reviewer X5No**
>
> We thank the reviewer for the thoughtful insights and comments.
>
> 1. >“The authors provide theoretical results comparing VariBAD (a meta-RL algorithm) with Dreamer (a model-based algorithm) on decomposable tasks which gives grounds for why such tasks are hard for model-free meta-RL algorithms.”
>
>    To clarify our findings: our theoretical results show that while a general high-dimensional task distribution is inherently hard for **any** meta-RL approach (see [1] and [2]), the special case of decomposable task distributions (which are also high dimensional) have better guarantees. Further, we showed that in order to achieve those guarantees one needs to use a local reconstruction approach instead of a global one. As a final clarification, our theory doesn’t compare between VariBAD and Dreamer in particular, and doesn’t  analyze the differences in guarantees between model-free and model-based approaches.
>
> 2. Thank you for pointing out these references. As mentioned in the general comment to all reviewers above, the field of meta-RL is rich and diverse, and we attempted to focus our related work section on what we deemed most relevant to our paper. For the sake of completeness, we have incorporated these additional papers into an extended related work section supplied in the appendix (Section F) of the revised version of the paper.
>
>    We also compared MAMBA with $RL^2$, and a recent related method, DREAM. Both experiments can be found in the additional experiments supplementary PDF, and demonstrate that MAMBA is preferable over these two algorithms as well.
>
> 3. We have followed previous work ($RL^2$, PEARL, VariBAD, HyperX) and focused our investigation on common meta-RL  benchmarks. We acknowledge the importance of standardized benchmarking in meta-RL; as it seems like Alchemy is a benchmark that is gaining traction, we will be happy to test MAMBA on it and report our results in a future version of the paper. That said, our current investigation already makes extensive comparisons between MAMBA and other meta-RL algorithms on well-established benchmarks.
>
> Finally, we thank the reviewer for the helpful wording and styling suggestions, we revised some of the text as suggested.
>
> [1] Mandi et al., 2022, On the effectiveness of fine-tuning versus meta-reinforcement learning
>
> [2] Rimon et al., 2022, Meta reinforcement learning with finite training tasks-a density estimation approach.

---

> > ### Comment · Reviewer_X5No · 2023-11-22
> > **Response to authors**
> >
> > I think that the authors for the careful reflections throughout. I have changed my score accordingly.

---

> > > ### Author Response · Authors · 2023-11-22
> > >
> > > Thank you for your quick reply, and for raising the review score. Please let us know if you have any further questions.

---

### Official Review · Reviewer_fKkT · 2023-11-01

**Soundness:** 3 good
**Presentation:** 3 good
**Contribution:** 3 good
**Rating:** 6
**Confidence:** 4

**Summary:**

This work draws a connection between model-based RL and a common meta-RL approach, VariBAD. Specifically, both encode the trajectory history into a latent variable, which is then used to predict observations, e.g., the state and rewards. This work leverages this connection to propose a new meta-RL approach based on Dreamer called MAMBA. Compared to Dreamer, MAMBA differs in three ways:
(1) Like many meta-RL algorithms, MAMBA augments the state with the current timestep and the reward.
(2) Whereas Dreamer only computes the latent variable over the past 64 steps of the history, MAMBA computes it over the entire past history. This helps in the cases where important information was discovered more than 64 steps ago.
(3) MAMBA introduces a curriculum where the horizon of the episode is increased over time to help combat the effect where long-range model predictions are inaccurate at the beginning of training.

This work evaluates MAMBA and finds that it performs favorably compared to VariBAD and Dreamer.

**Strengths:**

*Clarity*
- This work is well-written and easy to understand. The problem this paper is attempting to solve, and its proposed algorithm are clearly presented, which makes it easier to reproduce.

*Originality and Significance*
- The proposed changes appear to be fairly minor: augmenting the state with these additional observations is something that happens in a fair number of other papers; and increasing the history length for computing the context variable is akin to increasing a hyperparameter in Dreamer, which is reduced for computational reasons. However, they seem fairly sensible and result in fairly good performance gains.
- Additionally, this work makes interesting connections between model-based RL and VariBAD, and shows that Dreamer can outperform existing meta-RL algorithms.

Overall, this work unproblematically provides several contributions that are interesting to the meta-RL community, so I would be in favor of acceptance.

**Weaknesses:**

I think this work already provides valuable contributions, but can primarily be strengthened significantly by shedding more light on its results.
- MAMBA proposes 3 changes over Dreamer. It would be very helpful to perform an ablation study on these 3 changes and understand which ones are most important and how they impact performance.
- This work finds that Dreamer performs better than VariBAD and HyperX out-of-the-box on meta-RL tasks. I find the paper's claim that this performance gap probably results from architectural differences (VariBAD uses a very simple architecture, whereas Dreamer's is heavily tuned), but it would be interesting to ablate this and understand this better. At face value, non-meta-RL algorithms outperforming meta-RL algorithms on meta-RL tasks is somewhat surprising, and warrants further investigation. Further, it would be worth understanding in what ways these algorithms differ out-of-the-box. Does Dreamer result in better exploration? Or just exploitation? Computing the results in Figure 4 for VariBAD could be very helpful. Additionally, on what sorts of tasks do we expect Dreamer to be better than VariBAD? The tasks in this work take a very particular structure (they decompose nicely into iterated tasks). Is this a structure that clearly benefits one of the algorithms? What happens when this structure is violated?
- Further analysis of when it's important to predict the whole past + future (as in VariBAD) vs. local reconstruction (as in Dreamer) would be helpful. It's quite believable that local reconstruction could result in better empirical performance due to optimization, though the reason why VariBAD predicts the whole trajectory is as a proxy for predicting a distribution over dynamics, which is intractable. Predicting long-past events could in principle be really important for learning a good representation if some observation explains the dynamics seen early on, though this structure is not present in the current tasks, which may contribute to the conclusions drawn in this work. Some experiments or discussion on this would be helpful.
- Similarly, discussion about when it is necessary / possible to use the entire past to compute the context variable would be helpful. For really long horizon tasks, rolling out a recurrent policy is simply computationally intractable, so chunking is necessary, e.g., along the lines of R2D2 (https://openreview.net/pdf?id=r1lyTjAqYX)
- DREAM (https://arxiv.org/abs/2008.02790) reports better exploration than VariBAD on tasks requiring more sophisticated exploration. It
and subsequent papers (https://arxiv.org/pdf/2211.08802.pdf) also provide more complex tasks. Given the results of Dreamer outperforming VariBAD out-of-the-box, it would be interesting to consider discussion or comparison with DREAM or on these more complex tasks.
- Despite meta-RL being core to this work, the related works section glosses over a rich literature on meta-RL, including model-based meta-RL e.g., see https://arxiv.org/abs/2301.08028, https://arxiv.org/abs/1905.06424, https://arxiv.org/abs/1803.11347, https://arxiv.org/abs/1809.05214

These open questions are what justify my score of a 6 over an 8.

**Questions:**

Please see previous section.

---

> ### Author Response · Authors · 2023-11-21
> **Response to reviewer fKkT**
>
> We thank the reviewer for taking time to review our paper, and for their thoughtful suggestions and insights.
> We have added multiple experiments during the rebuttal period, the results of which are described in the general comment above and can be viewed in the rebuttal PDF (uploaded in the supplementary materials). Central results are described below:
>
> 1. **Ablation Study of MAMBA:**
>    * We first note that it is infeasible to run experiments without adding the reward into the observation because it prevents the agent any possibility to distinguish between tasks – so we cannot remove this feature in the meta-RL context; to validate this, we removed the reward observation and tested on the Rooms-3 environment. The results are as expected:
>
>    |                          |    **MAMBA**   | **MAMBA w/o Reward Observations** |
>    |--------------------------|:--------------:|:----------------------------------:|
>    | **Average Total Return** | 156.2$\pm$ 1.7 |            11.5$\pm$ 1.4           |
>
>    * We note that Mamba with the original batch sampling (i.e. batches do not start from the start of the meta-episode) is in fact the second baseline (Dreamer – our params).
>    * Finally, we’ve tested the effects of removing the horizon scheduling in the HalfCircle scenario, and indeed found that the results obtained are on-par but runtime is slower (on the same hardware):
>
>    |                          |    **MAMBA**   | **MAMBA w/o Horizon Scheduling** |
>    |--------------------------|:--------------:|:---------------------------------:|
>    | **Average Total Return** | 156.2$\pm$ 1.7 |           156.8$\pm$ 1.2          |
>    |    **Total Time [H]**    |      34.1      |                44.0               |
>
>    * Due to space constraints, we added these experiments to the appendix (Section G).
>
> 2. We compared to DREAM as suggested by the reviewer. We tested DREAM in the Rooms environment as this was the most similar to the Minigrid environments in the original paper. We found that DREAM was unable to produce a useful policy. See more details in the additional experiments PDF in the supplementary material.
>
> Next, we respond to the questions raised by the reviewer:
>
> 3. **Regarding the cause of the gap in performance between VariBAD and Dreamer:**
>     * Dreamer was originally designed for POMDPs which meta-RL is a flavor of, it is not very surprising that Dreamer is effective in this context. In addition, Dreamer can be seen as an effective out-of-the-box algorithm with many moving parts, and it is difficult to understand the individual contribution of each part. Moreover, it has been used as-is in many other settings with great success. Finally, a full comparison between Dreamer and VariBAD is detailed in Section 3.3 in our paper.
>     * **Regarding exploration/exploitation:** we have modified Figure 4, adding the final performance of VariBAD and HyperX. We note that MAMBA has both better exploration (as seen by the high rewards in the first episode of every meta-episode) and better exploitation (as seen by the higher total returns of the different sub-episodes).
>    * >“The tasks in this work take a very particular structure (they decompose nicely into iterated tasks). Is this a structure that clearly benefits one of the algorithms?”
>
>       As shown by our theory Section 3.2.1, a method with local reconstruction is expected to work better than one with global reconstruction on decomposable tasks. We also showed that while local reconstruction is beneficial for VariBAD as well, dreamer still outperforms it (Section 3.2.2).
>
>       >“What happens when this structure is violated?”
>
>       Our results show that Dreamer works better on a variety of task structures including non-decomposable meta-RL tasks. See our comment to this effect in the general response to all reviewers above.
>
> 4. **Regarding the local vs. global reconstruction:**
>    * In Section 3.2 of the paper we do analyze, both theoretically and empirically, when and why local reconstruction should be helpful (i.e decomposable tasks).
>    * There is a difference between encoding and reconstruction. Our work also encodes the whole history (from the beginning). Every meta-rl algorithm should do that in order to pass information from previous episodes to future ones.

---

> > ### Author Response · Authors · 2023-11-21
> > **Response to reviewer fKkT (cont.)**
> >
> > 5. **Regarding the context length limitation:**
> >    Every context-based meta-RL algorithm is limited in a similar manner. Indeed it would be interesting to explore chunking long trajectories for computational efficiency, while risking missing important signals. This could be a good avenue for future research.
> >
> > 6. **Regarding related work:** thank you for highlighting these additional papers. As mentioned in the general comment to all
> >    reviewers above, we aim to cover the most relevant literature in our related work section. However, as the field of meta-RL is rich with a variety of approaches, we acknowledge that we may have overlooked or underrepresented some of them. In accordance, we added an extended related work section (see Appendix F in the revised version of the paper) where we provide additional background, and elaborate further on papers already in our related work section.

---

> ### Comment · Reviewer_fKkT · 2023-11-23
>
> Thanks to the authors for the thorough response. Overall, I find the additional experiments and discussions to be useful and interesting. I remain in support of accepting the paper, though I am not willing to raise my score all the way to an 8 (the next possible score).
>
> - It’s worth noting that the most novel proposed change in this paper doesn’t seem to be extremely important for improved performance based on the authors new ablation studies. This is fine, but worth noting.
> - I am somewhat skeptical of the reported DREAM results. The rooms task is of similar horizon and arguably simpler complexity than the tasks considered in the DREAM paper. Hence, I find it surprising that the authors report that DREAM fails entirely. I find the authors’ explanation of hyperparameter tuning etc to be plausible, though the comparison seems somewhat sloppy if baselines aren’t tuned for the specific domain they’re being applied to. It also seems possible to be due to implementation error, since the DREAM paper itself and follow ups also suggest relative robustness to hyper parameters. I don’t particularly hold this against the authors and still support acceptance.

---

> > ### Author Response · Authors · 2023-11-23
> > **We thank the reviewer for the response**
> >
> > We thank the reviewer again for the time and effort of evaluating our paper.
> >
> > Regarding the first point: we think that the most crucial and novel feature of our proposed method is actually training the agent from the initial step in the meta-episode with the modifications this entails. The horizon scheduling is mostly helpful not to increase the returns, but rather to make the experiments feasible under low compute budgets. To further convey this point, we could ablate the horizon schedule on a harder task such as Reacher (where we expect that the overhead would be much apparent).
> >
> > Regarding the second point, we'd like to elaborate on the evaluation process for DREAM to make our assessment clearer: we used the original DREAM repository and hyper-parameters, inserting our Rooms environment into that code base (we thought this is valid comparison especially because the DREAM literature claims to be mostly robust to hyperparameter variations). As for the results, both DREAM and MAMBA start with returns ranging from -5 to -10, after 10M steps we see that DREAM manages to sometimes reach returns of  less than 2, while MAMBA reaches scores of around 155. To conclude, we note that this result actually demonstrates the most important attribute of MAMBA (at least in our view): namely, managing to solve new environments with out-of-the-box hyperparameters.
> > Nonetheless, we agree with the reviewer that it would be interesting to investigate further and figure out how to unlock the performance of DREAM on our environments.

---

### Official Review · Reviewer_cu12 · 2023-11-01

**Soundness:** 3 good
**Presentation:** 3 good
**Contribution:** 2 fair
**Rating:** 6
**Confidence:** 3

**Summary:**

The authors introduce MAMBA (MetA-RL Model-Based Algorithm), a meta-reinforcement learning algorithm developed using the Dreamer framework. The authors extend the Dreamer framework which was designed for POMDPs to the general case of meta-reinforcement learning.  Additionally, the authors introduce two novel environments specifically for meta-reinforcement learning with high-dimensional task distributions, which can be decomposed into lower-dimensional sub-tasks. They demonstrate the performance of their algorithm on these new environments as well as standard meta-RL environments.

**Strengths:**

1. The algorithm is sample efficient when compared to other meta-RL algorithms.

2. The authors conduct a good number of simulations to explain their algorithm, and evaluate its performance.

3. The paper is generally well written and easy to follow.

**Weaknesses:**

**The assumption of task decomposability and task independence is strong, vague, and confusing**

The paper assumes scenarios of task decomposability, where each task is decomposed into independent tasks, I think this is a pretty strong assumption, and not many environments will satisfy this criteria. The example quoted for task decomposability by the authors is a little confusing, the authors  provide an example of a robot being required to solve several independent problems in a sequence.  Isn't the fact that they should be performed in a sequence make the tasks dependent, and thus not independently decomposable? Further, in the Appendix (section C multi-goal Reacher-N) the authors claim that the environment is decomposable into *nearly independent sub-tasks*, which is a little vague and confusing.

**Algorithmic contribution is minimal**

The proposed algorithm is a minor modification over Dreamer, with the only significant change being sampling full-meta episodes instead of a smaller fixed length

**Questions:**

1. What determines the number of sub-episodes for each environment? It appears that each environment has a different number of sub-episodes. How does altering the number of sub-episodes influence the algorithm's performance?

2. The algorithms are tested in deterministic environments (although I could be mistaken). How do you anticipate the performance would be affected if the environments were stochastic? Could the introduction of randomness potentially complicate the task identification process?

**Minor Clarifications**

1. In Figure 1 (left most) Why isn't the agent exploring the right portion of the semi-circle during the first episode?

2. To enhance understanding, it would be beneficial to have comprehensive descriptions of the new environments (Multi Goal Rooms and Multi Goal Reacher), including details about their state-space, action space, and reward structure.

3. I would encourage the authors to add markers to all plots to enhance readability.

---

> ### Author Response · Authors · 2023-11-21
> **Response to reviewer cu12**
>
> Thank you for your time spent reviewing our work, and for your valuable insights.
>
> 1. **Regarding the task decomposability assumption:**
>
>    We would like to clarify that MAMBA does not assume task decomposability, and works well for many meta-RL tasks that are not decomposable. The focus on decomposable domains in the paper is for analysis purposes: for this class of problems, we can understand *why* our approach is more suitable. Obviously, real world problems are not necessarily decomposable, and we provide sufficient evidence that MAMBA is beneficial even without this structure in the task. See our response to this effect in the general comment to all reviewers above.
>
> 2. >“Isn't the fact that they should be performed in a sequence make the tasks dependent, and thus not independently decomposable?”
>
>    Thank you for this excellent question, you are correct – in this case, every subtask may affect the initial distribution of future sub-tasks. While this is true, our theoretical results still stand as they are not affected by the initial distributions of the sub-tasks. For completeness and clarity we modified our theory and discussion sections to refer to tasks which are decomposable into independent sub-tasks up to the initial distribution.
>
>    We further clarify that although the initial distribution is affected by previous tasks, the rewards and dynamics attributed to every sub-task are independent and are determined only according to the goals.
>
> 3. > “The proposed algorithm is a minor modification over Dreamer”
>
>    Correct. Our main contribution is showing that with relatively minor, yet tailored, modifications of Dreamer we achieve state-of-the-art results in meta-RL (with significant improvement over Dreamer-Vanilla). Moreover, we view the fact that we only needed minor modifications as a central contribution of this paper rather than a drawback, as one of our objectives is to show that Dreamer is successful in meta-RL settings as-is.
>
> 4. **Regarding choice of the number of sub-episodes:**
>
>    For environments borrowed from VariBAD \ HyperX, we use the same number of sub-episodes as in these works. In new environments, we choose the number of episodes to be sufficient to explore all goals in the goal distribution.
>
> 5. **Regarding stochastic environments:**
>
>    Indeed, all of the environments we have tested had deterministic dynamics and reward functions. However, there is no reason for stochasticity to make the task identification harder; to test this, we added an investigation of a variant of the HalfCircle environment, where a stochastic wind force is added at each step. We added the results to the supplementary experiments PDF and we also present them here for convenience:
>
> |                          |   **VariBAD**   |   **HyperX**   | **Dreamer-Vanilla** | **Dreamer-Tune** | **MAMBA (Ours)** |
> |--------------------------|:---------------:|:--------------:|:-------------------:|:----------------:|:----------------:|
> | **Average Total Return** | 194.5$\pm$ 44.7 | 177$\pm$ 118.5 |    226.1$\pm$3.5    |   224.5$\pm$4.4  |  224.1$\pm$5.2 |
> |      **Frames [M]**      |        20       |       20       |          2          |         2        |         2        |
>
>    As can be seen in the table, the return did not drop significantly from the original results in Table 1 of the paper, and the Dreamer variants are still superior to the baselines. All of the Dreamer variants perform close to Bayes-optimally. Some samples from the learned policy are presented in our [anonymous website](https://sites.google.com/view/mamba-iclr2024)
>
> 6. >“In Figure 1 (left most) why isn't the agent exploring the right portion of the semi-circle during the first episode?”
>
>    As can be seen in our [anonymous website](https://sites.google.com/view/mamba-iclr2024#h.c0hgsu9rnu35) in the far left example of the Point Robot section, after missing the goal the agent finds it in later episodes. It is likely that it is suboptimal to “waste time” exploring the far right of the half circle first as it will lower the return for the rest of the location for the off-chance the goal is actually there. Also, from the HyperX paper (figure 7c) we can see the same behavior appears in HyperX and VariBAD - the agent does not explore the far end of the half circle in the first exploration episode, suggesting that it is indeed suboptimal to do so.

---

> > ### Comment · Reviewer_cu12 · 2023-11-21
> >
> > I thank the authors for their rebuttal and clarifications. The modifications made to the main paper clarify some of the questions posed by other reviewers as well. I have modified my score accordingly.
> >
> > **Minor point:** I noticed a typo in the first line of Appendix G in the modified draft.

---

> > > ### Author Response · Authors · 2023-11-22
> > >
> > > We are glad we were able to alleviate your concerns. Thank you for replying promptly and for raising your score. We have uploaded a revision fixing the typo. Please let us know if you have any further questions or concerns.

---

### Author Response · Authors · 2023-11-21
**General Comment to all Reviewers and AC**

First, we thank the reviewers for their time and thoughtful suggestions for improvements to our paper. We address some recurring topics here, and dive further into the reviewers’ comments in the individual responses.

1. **Experiments – additional domains and baselines.**
Note that our hyper-parameter configuration remains fixed (except for the ratio of model updates per environment steps). Full results for these experiments can be found in the rebuttal PDF as well as videos uploaded as the supplementary materials. We summarize the results here:
    * DREAM [3]: Because the experimental settings of DREAM mostly focus on goal conditioned meta-RL tasks in GridWorld environments, we tested it in our Rooms environment. DREAM fails to obtain any meaningful policy. This may result from a mismatch in hyper-parameters between the original paper’s scenarios and Rooms, but this is also one of the advantages of using a Dreamer-based architecture like in MAMBA (same parameters work well on different scenarios out-of-the-box).
    * $RL^2$ [4]: this is a common meta-RL baseline; our additional experiments show that MAMBA is superior to $RL^2$, except in EscapeRoom, where the difference is slightly in favor of $RL^2$ (although $RL^2$ used 5x more data than MAMBA in this environment).
    * Harder environments:
        * Rooms7-8: We added more rooms to challenge MAMBA. We saw that MAMBA still manages to find a Bayes-optimal behavior and the drop in return is attributed to the exploration required to find the additional goals.
        * PandaReach [1]: This is a complex system with visual inputs and sparse rewards that requires a non-trivial Bayes-adaptive policy. In this environment a 7-DoF robot arm is required to reach goals with the robot end-effector. The goals are hidden and must be discovered through exploration. The rewards are sparse, i.e.  only when the end-effector is within a short distance of the goal, a reward of 1 is obtained (otherwise the reward is 0). The observation is an image of size 84x84x4, and the action space is the direction the end-effector should go towards. After 6M steps for Dreamer-Vanilla, Dreamer-Tune, and MAMBA, the returns are 5.3, 14.5, and 133.3 respectively, showing again that MAMBA is better than the other Dreamer versions. Also note that [1] reported return of ~135 in this environment (results were given as plots) after 20M environment steps compared to our 6M steps. Performance of our approach after 20M steps will be included for the final version.
        * More common environments: We add experiments on the Humanoid-Dir-2D domain, described in Rakelly et al. [2]. In this environment, the Humanoid MuJoCo task is initialized with a random target walking direction $\theta \in [0, 2\pi]$ on the 2D plane. The results indicate that in this scenario, using less environment time-steps (30M vs. 100M), the algorithms based on the Dreamer architecture are better than VariBAD and HyperX, and among those MAMBA is substantially better, reaching the highest return.
        * HalfCircle-Wind: HalfCircle with stochastic dynamics as suggested by reviewer cu12. The same conclusions for the original HalfCircle also hold here.

2. **Some reviewers have commented on the notion of task decomposability**. We’d like to make an important clarification: MAMBA is not limited to decomposable tasks, and works well on a variety of meta-RL baselines, as we show in many of the results in the paper (see Reacher-1, HalfCircle, HalfCircle-Wind, Escape Room and Humanoid-Dir). While we highlight decomposability and explain why MAMBA is expected to outperform baselines on these tasks, our results on other types of tasks clearly show that MAMBA is not limited to these settings.

3. **Multiple reviewers have suggested additional related papers for comparison and consideration**. First of all, we’d like to thank the reviewers for spending the time to find and cite these papers. As the reviewers are well aware, meta-RL is a rich and fast-growing field of study. We believe we have made an honest effort to include the most relevant related work in our paper, considering space constraints. However, we acknowledge that we may have overlooked some approaches, and we therefore add an extended related work section to our revised paper (see Appendix F).

In addition to this comment, the rebuttal PDF and the replies to each individual reviewer below, we have uploaded a revised version of the paper conforming with the various comments and new results. Changes from the original submission are marked in blue.

[1] Choshen and Tamar, Contrabar: Contrastive bayes-adaptive deep rl. ICML, 2023.

[2] Rakelly et al., Efficient off-policy meta-reinforcement learning via probabilistic context variables. ICML, 2019.

[3] Liu et al., Decoupling Exploration and Exploitation for Meta-Reinforcement Learning without Sacrifices, ICML 2021.

[4] Duan et al., $RL^2$: Fast Reinforcement Learning via Slow Reinforcement Learning, ICLR 2017

---

### Comment · Area_Chair_2q6j · 2023-11-23
**Author-Reviewer discussion period ending *very* soon**

Thank you to the reviewers for responding. The authors have put great effort into their response, so can I please urge the other reviewers (fKkT, tjN7) to answer the rebuttal.
Thank you!

---

### Meta-Review · Area_Chair_2q6j · 2023-12-05

**Metareview:**

The paper introduces a model-based meta-reinforcement learning (meta-RL) approach leveraging successful elements from existing methods. It addresses low sample efficiency in meta-RL by achieving higher returns with improved efficiency, requiring minimal hyperparameter tuning. The approach is validated across common benchmark domains and shows promise in more complex, higher-dimensional environments. After a fruitful rebuttal period, all 4 reviewers came to a consensus of accepting the paper.

**Justification For Why Not Higher Score:**

No reviewers championed the paper heavily to be accepted; all voted that the paper marginally crosses the line for acceptance.

**Justification For Why Not Lower Score:**

N/A

---

### Decision · Program_Chairs · 2024-01-16

Accept (poster)